# Evaluating the Reversal Curse in Model Editing

**Hao-Xiang Xu**[*]                                                    *nh2001620@mail.ustc.edu.cn*
*National Engineering Research Center of Speech and Language Information Processing*
*University of Science and Technology of China*

**Jun-Yu Ma**[*]                                                       *mjy1999@mail.ustc.edu.cn*
*National Engineering Research Center of Speech and Language Information Processing*
*University of Science and Technology of China*

**Zhen-Hua Ling**                                                      *zhling@ustc.edu.cn*
*National Engineering Research Center of Speech and Language Information Processing*
*University of Science and Technology of China*

**Quan Liu**                                                           *quanliu@iflytek.com*
*iFLYTEK Research*

**Cong Liu**                                                           *congliu2@iflytek.com*
*iFLYTEK Research*

**Jia-Chen Gu**[†]                                                     *gujc@ucla.edu*
*University of California, Los Angeles*

**Reviewed on OpenReview:** *https://openreview.net/forum?id=jAHwodCUxP*

## Abstract

Large language models (LLMs) are prone to hallucinate unintended text due to false or outdated knowledge. Since retraining LLMs is resource intensive, there has been a growing interest in *model editing*. Despite the emergence of benchmarks and approaches, existing *unidirectional* editing and evaluation paradigms have failed to explore the *reversal curse*. In this paper, we study bidirectional language model editing, aiming to provide a rigorous evaluation to assess if edited LLMs can recall the editing knowledge bidirectionally. A metric of *reverse generalization* is introduced and a benchmark dubbed **B**idirectional **A**ssessment for **K**nowledge **E**diting (BAKE) is constructed to evaluate if post-edited models can recall the edited knowledge in the reverse direction of editing. We conduct extensive experiments using a variety of editing methods and LLMs. The results show that while most editing methods are able to accurately recall editing facts along the modification direction, they exhibit substantial systematic deficiencies when evaluating in the reverse direction. To further investigate the underlying causes of reversal curse and to explore potential strategies for mitigation, a detailed analysis is conducted from three perspectives. Our findings reveal that although In-Context Learning (ICL) can mitigate the reversal curse to a certain extent, it lacks continuity, is limited by the input length, and may introduce hallucinations. Therefore, combining the advantages of ICL and other editing methods is a promising direction for developing new editing paradigms.

## 1 Introduction

Large language models (LLMs) have shown impressive capabilities in understanding and generating text, improving performance on a variety of tasks (Chen et al., 2021; Ouyang et al., 2022; Zhang et al., 2023a).

---

[*]Equal contribution.
[†]Corresponding author.

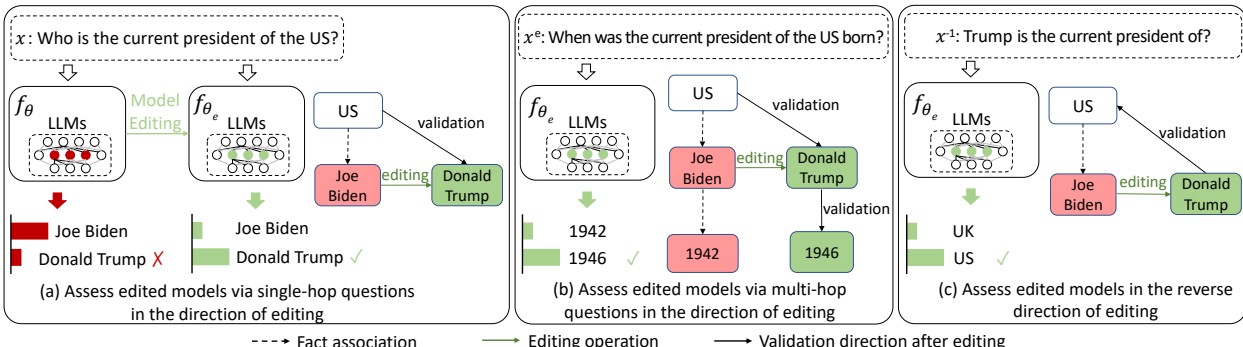

Figure 1: Comparison of *unidirectional* evaluation paradigms that assess whether edited models can recall the editing facts (a) via single-hop questions, or (b) via entailed questions in the direction of editing; and (c) the proposed *bidirectional* paradigm that assesses model editing in the reverse direction of editing. $f_\theta$ / $f_{\theta_e}$ denotes the models before / after editing.

However, existing LLMs inevitably exhibit hallucinations due to incorrect or outdated knowledge stored in their parameters (Zhang et al., 2023b; Peng et al., 2023; Ji et al., 2023). Considering the huge computing resource consumption of retraining LLMs and echoing green deep learning (Schwartz et al., 2020; Xu et al., 2021), there has been considerable and growing attention on the research regarding *knowledge editing* (a.k.a., *model editing*) (Sinitsin et al., 2020; Cao et al., 2021; Meng et al., 2022; 2023; Dai et al., 2022; Mitchell et al., 2022b; Yao et al., 2023; Gu et al., 2024; Xu et al., 2025), which aims at efficiently modifying and updating specific knowledge without full retraining.

Existing model editing methods fall into two categories (Yao et al., 2023): *Parameter-preserving methods* (Wang et al., 2024a), and *Parameter-modifying methods* (Mitchell et al., 2022b; Meng et al., 2023). The effectiveness of an editing method is usually evaluated along three dimensions of *efficacy*, *generalization* and *locality*. The ZsRE (Levy et al., 2017) and CounterFact (Meng et al., 2022) datasets are primarily utilized to assess whether edited models can recall the newly injected facts, as depicted in Figure 1(a). Besides, Yao et al. (2023) and Zhong et al. (2023) have introduced *portability* and MQuAKE respectively to extend the assessment of generalization by measuring whether edited models can correctly answer the questions entailed by the editing facts, as shown in Figure 1(b). Despite the emergence of benchmarks and techniques for editing, they all operate under the *unidirectional* paradigm following only the direction being edited. The *reversal curse* (Berglund et al.) has not yet been explored in model editing, and it remains unclear whether edited models can reason and recall the editing facts in the reverse direction of editing, as shown in Figure 1(c).

In this paper, we study bidirectional language model editing, determining if edited LLMs can accurately recall the editing knowledge bidirectionally. Among various kinds of learned beliefs such as logical, spatial, or numerical knowledge, this paper investigates factual knowledge within LLMs, where each piece of knowledge is represented in the form of a *(subject, relation, object)* triple. A metric of *reverse generalization* is introduced to evaluate the effectiveness of edited models in recalling knowledge in the reverse direction of editing. A benchmark dubbed **B**idirectional **A**ssessment for **K**nowledge **E**diting (BAKE) is constructed. When inverting the relation between subject and object, we adopt a taxonomy that categorizes relations between entities into four classes: *one-to-one*, *one-to-many*, *many-to-one*, *many-to-many* (Bordes et al., 2013), is considered. To characterize these symmetrical and asymmetrical relations, two evaluation forms of *question answering* (QA) and *judgment* are adopted for evaluation.

To establish the comparison of existing model editing methods on bidirectional model editing, this paper conducts meticulous experiments on nine editing methods in two categories, including parameter-modifying methods: FT (Zhu et al., 2020), MEND (Mitchell et al., 2022a), MEMIT (Meng et al., 2023) ROME (Meng et al., 2022), RECT (Gu et al., 2024), AlphaEdit (Fang et al., 2025), and parameter-preserving methods: GRACE (Hartvigsen et al., 2023), WISE (Wang et al., 2024a) and IKE (Zheng et al., 2023). Five representative LLMs of varying sizes are selected as the base models including GPT-2 XL (1.5B) (Radford et al., 2019), GPT-J (6B) (Wang & Komatsuzaki, 2021), LLaMA-2 (7B) (Touvron et al., 2023), LLaMA-2 (13B) (Touvron

et al., 2023) and LLaMA-3 (8B) (Dubey et al., 2024). We surprisingly observe that the vast majority of current editing methods and LLMs, while effective in recalling editing facts in the direction of editing, suffer serious deficiencies when evaluated in the reverse direction. Strikingly, the LLaMA-3 (8B) model edited by the state-of-the-art AlphaEdit (Fang et al., 2025) can recall 98.45% of the editing facts in the editing direction, but only 0.88% of the editing facts in the reverse direction. Another interesting finding is that when an in-context learning based method IKE is used for model editing, the reverse knowledge recall accuracy on LLaMA-3 (8B) reaches 51.78%, indicating that the reversal curse could be mitigated to some extent.

To further investigate the underlying causes of the reversal curse, a detailed analysis is conducted from three perspectives. Our findings and conclusions are mainly in three aspects: (1) We compared the probability of the model generating the original answer and the desired answer after editing. Unexpectedly, the models edited by most methods did not increase the probability of the desired answer, which indicates that they lack a true understanding of editing knowledge, leading to the "reversal curse". (2) The ICL mechanism can help other editing methods mitigate the reversal curse to some extent, but it lacks continuity and is limited by the length of the context window. (3) When using ICL-based methods for editing, the model may output answers that are irrelevant to the question. Although these findings reveal that the ICL mechanism can mitigate the reversal curse to a certain extent, it lacks continuity, scalability, and may introduce hallucinations. Therefore, it is particularly important to better combine it with existing editing methods to develop new editing paradigms.

In summary, our contributions are three-fold: (1) This study makes the first attempt to explore bidirectional language model editing. A metric of reverse generalization is introduced and a benchmark of BAKE is constructed to assess the reverse generalization of edited models. (2) This paper conducts meticulous experiments and presents surprising findings that existing methods and LLMs suffer from the severe reversal curse on model editing. (3) This paper presents a detailed analysis and provides initial insight into the underlying causes of the reversal curse, while also exploring a potential direction to mitigate the problem. All data and code are available to facilitate reproducing our results. We hope that both our benchmark and analysis can shed light on bidirectional language model editing.

## 2 Related Work

**Model Editing**   Existing model editing methods can be broadly categorized into parameter-modifying and parameter-preserving approaches. *Parameter-modifying methods* inject new knowledge by directly updating model weights within the original architecture, including meta-learning approaches such as KE (Cao et al., 2021), MEND (Mitchell et al., 2022a), and InstructEdit (Zhang et al., 2024), as well as locate-then-edit methods like ROME (Meng et al., 2022), MEMIT (Meng et al., 2023), and EAC (Xu et al., 2025), which utilize causal tracing and compressed updates to reduce side effects. PRUNE (Ma et al., 2025) further localizes parameter changes by restricting the conditional number. *Parameter-preserving methods* retain original model weights by leveraging external modules or non-invasive strategies. IKE (Zheng et al., 2023) and DeCK (Bi et al., 2024) use in-context learning, SERAC (Mitchell et al., 2022b) employs external memory, T-Patcher (Huang et al., 2023) and CaliNet (Dong et al., 2022) inject additional neurons, GRACE (Hartvigsen et al., 2023) replaces hidden states with codebook representations, and WISE (Wang et al., 2024a) integrates updates via parameterized memory. Meanwhile, several benchmarks have been proposed to evaluate editing performance and reliability (Meng et al., 2022; 2023; Yao et al., 2023; Mitchell et al., 2022b; Zhong et al., 2023). For example, Zhong et al. (2023) use multi-hop questions to assess whether edits propagate consistently across related knowledge and datasets such as ConceptEdit (Wang et al., 2024c) focus on concept-level editing by modifying abstract definitions and semantic categories, providing a complementary perspective beyond entity-level factual editing. In addition, recent benchmarks derived from WikiData, such as TemporalWiki (Jang et al., 2022), WikiFactDiff (Khodja et al., 2024), and WikiBigEdit (Thede et al., 2025), focus on evaluating models' ability to update factual knowledge under temporal changes or large-scale editing scenarios, primarily from a forward editing perspective.

**The Reversal Curse of LLMs**   Recent works have studied the reversal curse (Berglund et al.; Lv et al., 2023; Guo et al., 2024; Lu et al., 2024; Apaolaza et al., 2025) in auto-regressive LLMs. Berglund et al. have verified that a model fine-tuned on a sentence of the form "*A is B*", it will not automatically generalize to the

reverse direction "*B is A*". Moreover, the likelihood of the correct answer will not be higher than for a random answer. This is termed the *reversal curse*. Contemporary to this work, Grosse et al. (2023) determined how much adding a given training example influences an LLM's outputs. For instance, given the input A, what most influences the likelihood of B? In their experiments, training examples that match the order ("*A precedes B*") are far more influential than examples with reverse order ("*B precedes A*"). Results show that if a pretrained model was not trained on facts in both directions, it would not generalize to both directions.

Compared with previous studies (Dai et al., 2022; Meng et al., 2022; 2023; Mitchell et al., 2022a; Yao et al., 2023; Zhong et al., 2023; Berglund et al.) that are the most relevant to our work, a main difference should be highlighted. These studies target only unidirectional model editing, while this study explores bidirectional editing and evaluation. Moreover, our analysis is conducted under a single-edit setting, aiming to isolate the reversal behavior of editing, rather than considering temporal or sequential update scenarios. To the best of our knowledge, this paper makes the first attempt to introduce a metric of *reverse generalization*, build a benchmark for evaluating the reverse generalization of editing methods, analyze the underlying causes of the reversal curse in model editing and point out future exploration directions to mitigate it.

## 3 Preliminary

The goal of model editing is to insert new facts into model parameters without retraining. In this paper, we study facts of the form $(s, r, o)$, consisting of a subject $s$, a relation $r$, and an object $o$ (e.g., $s =$ Eiffel Tower, $r =$ located in, $o =$ Paris). Following previous works (Zhong et al., 2023; Yao et al., 2023) to employ discrete prompts (question or cloze-style statement) to test whether a fact is stored in a model. Editing a fact is to insert a new triple $(s, r, o^*)$ in place of the current triple $(s, r, o)$, where these two triples share the same subject and relation. An editing operation is represented as $e = (s, r, o, o^*)$ for brevity.

Given an edit $e$ and a model $f$, model editing involves learning a function $K$ that yields an edited language model $f^* : K(f, e) = f^*$. To evaluate the effectiveness of editing methods, previous works focus on evaluating along three dimensions: efficacy, generalization and locality (Mitchell et al., 2022a; Meng et al., 2022).

**Efficacy** validates whether the edited models could recall the exact editing fact when presented with the sole editing prompt $p$. The assessment is based on Efficacy Score **(ES)** defined as: $\mathbb{1}[\arg\max_o P_{f^*}(o \mid p) = o^*]$, where $\mathbb{1}$ is the indicator function.

**Generalization** verifies whether the edited models could recall the edited fact under paraphrase prompts via Generalization Score **(GS)**: $\mathbb{E}_{p \in \mathcal{P}^G}[\mathbb{1}[\arg\max_o P_{f^*}(o \mid p) = o^*]]$.

**Locality**[1] is to verify whether the output of the edited models for inputs out of editing scope remains unchanged after editing under locality prompts via Locality Score **(LS)**: $\mathbb{E}_{p_l \in \mathcal{P}^L}[\mathbb{1}[\arg\max_o P_{f^*}(o \mid p_l) = o_l]]$, where $o_l$ was the original answer of $p_l$.

For example, given an editing fact ($s =$ Eiffel Tower, $r =$ located in, $o =$ Paris, $o^* =$ London), the editing prompt $p$ could be "Eiffel Tower is located in London". A paraphrase prompt could be "What is the location of the Eiffel Tower?". A locality prompt and its original answer could be "Where is the Louvre?" and "Paris".

## 4 BAKE: Bidirectional Assessment for Knowledge Editing

The BAKE benchmark comprises two datasets of BAKE-Q&J and BAKE-J. Since it is difficult to directly measure the outdated and false knowledge in LLMs, we follow previous works (Meng et al., 2022; Zhong et al., 2023; Yao et al., 2023), both datasets are designed for evaluating counterfactual edits in LLMs. When inverting the relation between subject and object, two evaluation forms, *question answering (Q)* and *judgment (J)* are adopted to characterize these symmetrical and asymmetrical relations.

---

[1]Also referred to as *specificity* in Meng et al. (2022; 2023).

### 4.1 Relation Category

There are various relations between entities in factual knowledge, which could be classified into four classes: *one-to-one*, *one-to-many*, *many-to-one*, and *many-to-many* (Bordes et al., 2013), as shown in Figure 2.

For *one-to-one* and *one-to-many* relations, both *question answering* and *judgment* forms are employed since there is a unique output after inverting the relation. Take "*developed*" relation from *one-to-many* category in Figure 2(b) as an example, one can construct the factual statement "*Apple developed iPhone*". When evaluated in the reverse direction, it could be presented to models in QA form, such as "*iPhone is developed by ___*" or *judgment* form like "*Whether iPhone is developed by Apple?*" Similarly, *one-to-one* category shares the same property.

On the other hand, only the *judgment* form is employed for *many-to-one* and *many-to-many* relations, since there are alternative outputs to a given reversal relation. Suppose a "*written by*" relation from *many-to-one* category shown in Figure 2(c) is chosen to form a statement "*The Trial is written by Kafka*". If presented to models for reverse validation in QA form like "*Kafka has written ___*", there may be many correct answers such as *The Metamorphosis* and *The Castle*. It is notable that edited models are expected to output The Trial as the intended answer for reversal evaluation instead of *The Metamorphosis* and *The Castle*. Therefore, evaluating the reversal relation for *many-to-one* and *many-to-many* categories via *judgment* is better. Similarly, the *many-to-many* category shares the same property as the *many-to-one*.

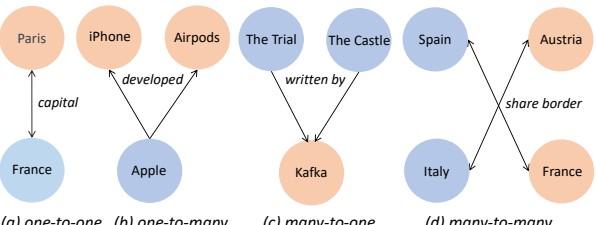

*(a) one-to-one*   *(b) one-to-many*   *(c) many-to-one*   *(d) many-to-many*

Figure 2: Examples of four relation categories. A given relation is considered (a) *one-to-one* if a subject can be associated with at most one object; (b) *one-to-many* if a subject can be associated with multiple objects; (c) *many-to-one* if multiple subjects can be linked to the same object; or (d) *many-to-many* if multiple subjects can be linked to multiple objects.

### 4.2 Data Construction of BAKE-Q&J

This dataset is constructed based on Wikidata (Vrandecic & Krötzsch, 2014), a knowledge base containing millions of fact triples. First, we manually chose a total of 25 *one-to-one* and *one-to-many* relations, and all fact triples under each relation, denoted as $F(r)$ = $\{(s, r, o) \mid s$ and $o$ are linked by $r$ in Wikidata$\}$. For each relation $r$, ChatGPT (gpt-4o) was used to generate templates $T(r)$ and corresponding inverse templates $T(r^{-1})^2$ $r^{-1}$. Appendix A provides some examples of these templates. Entities are substituted to form full prompts[3]: $\mathcal{P}(s, r) = \{t.\texttt{format}(s) \mid t \in T(r)\}$, where .$\texttt{format}()$ is string substitution. For example, a template for (r = written by) might be "*{} is written by*", where "*Hamlet*" substitutes "*{}*".

**Constructing counterfactual edits** To construct an editing example, two triples sharing the relation $r$ are sampled from Wikidata, denoted as $(s, r, o)$ and $(s^*, r, o^*)$ respectively. An edit is represented as $e = \{s, r, o, o^*, \mathcal{P}\}$ to test *efficacy*, where the editing prompt is denoted as $\mathcal{P}$. To test *generalization*, a set of two semantically-equivalent paraphrase prompts is sampled from $\mathcal{P}(s, r)\backslash \{\mathcal{P}\}$. Moreover, to test *locality*, a set of triples that share the same object $o$ are

Table 1: An example in the BAKE-Q&J dataset. $a$ and $a^*$ in different columns represent the original answer and desired answer after editing respectively.

| | |
|---|---|
| $\mathcal{E}$ | $(s, r, o)$: (Paris, capital of, France)
$(s^*, r, o^*)$: (London, capital of, England)
(Paris, capital of, France → England)
$p$: Paris is the capital of ($a$: France, $a^*$: England) |
| $\mathcal{P}^G$ | Paris, which is the capital of
Paris is the political center of which country?
($a$: France, $a^*$: England) |
| $\mathcal{P}^L$ | Palace of Versailles is located in
Marseille is the city of which country?
($a$: France, $a^*$: France) |
| $\mathcal{P}^{Rq}$ | What is the capital of England?
($a$: London, $a^*$: Paris) |
| $\mathcal{P}^{Rj}$ | Whether the capital of England is Paris?
($a$: no, $a^*$: yes) |

---

[2]Inverse relation $r^{-1}$ and its templates are only for evaluation.
[3]They could be in the form of question or cloze-style statement.

collected: $\mathcal{S} = \{(s', r', o)\}$. For example, select ($s=$ *Eiffel Tower*, $r=$ *located in*, $o=$ *Paris*), $\mathcal{S}$ might contain triple such as ($s=$ *France*, $r=$ *has capital*, $o=$ *Paris*). Then a set of prompts $\{\mathcal{P}(s', r') \mid (s', r', o) \in \mathcal{S}\}$ are constructed to sample the locality prompts $\mathcal{P}^L$.

**Constructing reverse prompts**   As for the *reverse generalization* proposed in this paper, after modifying the knowledge from $(s, r, o)$ to $(s, r, o^*)$, the edited model should be capable of deducing $(o^*, r^{-1}, s)$ if it is reversible. To ensure that the evaluation is valid after editing, we ensure that the triple $(o^*, r^{-1}, s)$ is not included (editing knowledge should not pre-exist), while $(o^*, r^{-1}, s^*)$ is included (original knowledge should pre-exist) in the original LLM. If not, this editing example will be filtered out accordingly. Specifically, given $o^*$ and $r^{-1}$, we tested if a model can output $s^*$ rather than $s$, to determine whether the triple $(o^*, r^{-1}, s^*)$ is included in the model. Furthermore, there are QA and *judgment* forms of prompts in this dataset and the reverse-qa prompt is defined as $\mathcal{P}^{Rq} = \{o^*, r^{-1}, s^*, s, p^{-1}\}$, where $p^{-1} \sim \mathcal{P}(o^*, r^{-1})$ and $s$ is the desired answer after editing (e.g., if $s=$Apple, $s^*=$Microsoft, $r=$developed, $o^*=$Windows, then $r^{-1}=$developed by, and $p^{-1}$ might be "*Windows is developed by which company?*"). Besides, the reverse-judge prompt $\mathcal{P}^{Rj}$ is a *judgment* question based on $p^{-1}$, for instance, "*Whether Windows is developed by Apple?*", which should be answered with "*yes*" after editing or "*no*" before editing.

## 4.3   Data Construction of BAKE-J

The construction of this dataset is similar to BAKE-Q&J, and the difference lies in that 20 *many-to-one* and *many-to-many* relations in total were manually selected. Moreover, only the *judgment* form is adopted for validating the reverse generalization, which has been explained in Section 4.1.

## 4.4   Dataset Summary

**Dataset Format**   As shown in Table 1, each example in the BAKE benchmark is represented as a tuple ($\mathcal{E}$, $\mathcal{P}^G$, $\mathcal{P}^L$, $\mathcal{P}^{Rq}$, $\mathcal{P}^{Rj}$), where $\mathcal{E}$ is the editing knowledge which will be injected into the language model. $\mathcal{P}^G$ and $\mathcal{P}^L$ are the prompts utilized to validate generalization and locality respectively; $\mathcal{P}^{Rq}$ and $\mathcal{P}^{Rj}$ correspondingly represent the reverse-qa and reverse-judge prompts for reverse generalization; $a$ and $a^*$ denote the original answer and desired answer after editing respectively. More examples can be found in Appendix B.1

**Dataset Statistics**   Readers can refer to Appendix B.2 for the details of the statistics of the BAKE-Q&J and BAKE-J datasets. To verify generalization and locality, there is at least one prompt for each editing example. Meanwhile, there is one reverse-qa prompt (only for BAKE-Q&J) and one reverse-judge prompt to verify reverse generalization. Both datasets consist of counterfactual edits utilized to investigate if edited models can recall the editing facts in both the editing and reverse directions.

# 5   Experiments

## 5.1   Experimental Setup

**Base LLMs**   Given limited computational resources, experiments were conducted on six LLMs of different sizes including **GPT-2 XL** (1.5B) (Radford et al., 2019), **GPT-J** (6B) (Wang & Komatsuzaki, 2021), **LLaMA-2** (7B) (Touvron et al., 2023), **LLaMA-3** (8B) (Dubey et al., 2024), **LLaMA-2** (13B) (Touvron et al., 2023) and **Qwen-3** (8B) (Yang et al., 2025).

**Editing Methods**   Nine model editing methods were selected as baselines, as shown in Table 2. Among them, IKE leverages in-context learning by injecting edited knowledge into demonstration examples, where relevant examples are retrieved based on semantic similarity and organized to guide the model's predictions without modifying model parameters. Readers can refer to Appendix C.1 for more details.

**Evaluation Metrics**   In addition to the proposed *Reverse Generalization*, three basic metrics of *Efficacy*, *Generalization* and *Locality* (Meng et al., 2022; 2023) introduced in Section 3 were adopted. Given two facts of $(s, r, o)$ and $(s^*, r, o^*)$, one editing operation updated $(s, r, o)$ to $(s, r, o^*)$ and the edited model $f^*$ was obtained.

Table 2: The categories of editing methods.

| Type | Parameter-modifying | | | | | | Parameter-preserving | | |
| | Gradient-based | | Locate-then-edit | | | | External parameter | | ICL-based |
| Method | FT | MEND | ROME | MEMIT | RECT | AlphaEdit | GRACE | WISE | IKE |

And the reverse-qa prompts $\mathcal{P}^{Rq}$ were formulated by $o^*$ and $r^{-1}$, with answer $s$. *Reverse Generalization* was to evaluate the effectiveness of edited models in recalling the editing knowledge under reverse prompts. For QA form prompts, Reverse-QA Score **(RQS)** was defined as: $\mathbb{E}_{p \in \mathcal{P}^{Rq}}[\mathbb{1}[\operatorname{argmax}_o P_{f^*}(o\,|\,p) = s]]$. As for judgment form prompts, Reverse-Judgment Score **(RJS)** was defined as: $\mathbb{E}_{p \in \mathcal{P}^{Rj}}[\mathbb{1}[\operatorname{argmax}_o P_{f^*}(o\,|\,p) = yes]]$. For all the above metrics, the average results across all edits in each dataset were reported. Define RS was as the average of RQS and RJS[4], the harmonic mean of ES, GS, LS, and RS was reported as Score **(S)** to comprehensively evaluate the performance of edited models in a more balanced manner.

## 5.2 Evaluation Results

Table 3 reported the results of different editing methods on BAKE-Q&J and BAKE-J. The RQS and RJS results for each base LLM were 0, since the counterfactual editing triples that pre-existed in each LLM have been filtered out as mentioned in Section 4.2.

**Existing methods perform well in the direction of editing** Most methods have high efficacy and generalization performance, showing that existing editing methods are capable of effectively recalling the editing facts in editing direction. However, the performance in terms of locality was still limited and deserves further investigation. Moreover, with the increase in model size, the performance of a specific editing method continued to improve on these three metrics. For instance, the performance of MEMIT applied to LLaMA-2 (13B) was notably higher than that applied to GPT-2 XL in terms of ES, GS, and LS by 23.33%, 27.91%, and 6.65% respectively on the BAKE-Q&J dataset.

**Most methods fail in the reverse direction of editing** Surprisingly, we observed that most editing methods and LLMs failed catastrophically when evaluated in the reverse direction of editing, especially in the QA form. Taking LLaMA-3 (8B) edited by AlphaEdit on BAKE-Q&J as an example, it achieved 98.45% (ES) which exhibited an impressive ability of recalling the editing facts in the direction of editing. However, it could only recall 0.88% (**RQS**) of the editing facts when evaluated in the reverse direction of editing using the QA format. Even when evaluated via the judgment form, it answered only 30.24% (**RJS**) of the questions correctly. These results show a surprising conclusion that, although current editing methods and LLMs can effectively recall the editing facts in the direction of editing, they suffer serious deficiencies when evaluated in the reverse direction. Our results suggest that current editing methods are far from fully understanding the injected knowledge, but just simply memorize via hard encoding. These also echo the *reversal curse* in Berglund et al. and Grosse et al. (2023), and we hope to attract more attention to this challenging issue.

**Different categories of editing methods have significant performance differences for the same metric** For RJS, the performance of gradient-based methods (FT, MEND) was markedly lower compared to methods that rely on locating knowledge neurons (MEMIT, ROME, AlphaEdit). This suggests that gradient updates demonstrate a myopic nature, showing a weak capacity to update information in the reverse direction. In addition, we found that the parameter-preserving methods (GRACE, WISE, IKE) outperformed the parameter-modifying approaches (RECT, AlphaEdit) on both LS and RJS metrics. This suggests that directly modifying model parameters introduces changes that hinder the model's ability to retain original knowledge and update information in reverse direction.

**IKE mitigates the reversal curse to some extent** As shown in Table 3, IKE was also applied to five LLMs. Specifically, under the same base model, IKE's ability to recall editing facts in the forward direction was comparable to that of other methods, with a slight improvement. More importantly, when

---

[4]Since there is no RQS in BAKE-J, only RJS is used to compute the harmonic mean.

Table 3: Evaluation results (%) of the BAKE-Q&J and BAKE-J datasets. The settings for these methods follow the published literature. The results of GPT-J and LLaMA-2 (7B) were placed in Appendix D. Best results in each column are highlighted in blue.

| Editor | BAKE-Q&J | | | | | | BAKE-J | | | | |
|---|---|---|---|---|---|---|---|---|---|---|---|
| | S | ES | GS | LS | RQS | RJS | S | ES | GS | LS | RJS |
| **GPT-2 XL (1.5B)** | | | | | | | | | | | |
| FT | 19.40 | 78.11 | 49.64 | 64.12 | 2.79 | 9.89 | 25.54 | 85.88 | 52.97 | 65.01 | 9.03 |
| MEND | 3.88 | 94.01 | 78.12 | 28.14 | 1.75 | 0.31 | 0.28 | 90.89 | 74.25 | 29.17 | 0.09 |
| MEMIT | 16.41 | 72.59 | 63.97 | 64.13 | 0.31 | 9.75 | 18.93 | 83.12 | 69.53 | 66.29 | 5.89 |
| ROME | 41.35 | 97.99 | 91.74 | 51.97 | 3.56 | 31.88 | 49.89 | 98.16 | 86.31 | 62.79 | 23.54 |
| RECT | 38.94 | 95.69 | 93.27 | 58.17 | 2.98 | 28.10 | 48.25 | 96.63 | 84.56 | 69.85 | 21.55 |
| AlphaEdit | 43.62 | 93.99 | 94.74 | 79.16 | 3.01 | 31.55 | 47.90 | 97.84 | 82.39 | 75.44 | 20.88 |
| GRACE | 38.51 | 88.05 | 30.84 | 99.99 | 4.58 | 35.35 | 43.78 | 91.60 | 35.52 | 95.98 | 23.88 |
| WISE | 50.63 | 92.87 | 90.04 | 99.68 | 5.12 | 37.34 | 55.07 | 90.69 | 85.61 | 98.78 | 25.12 |
| IKE | 78.26 | 92.45 | 95.19 | 89.97 | 38.14 | 68.94 | 84.70 | 98.34 | 88.43 | 85.37 | 71.26 |
| **LLaMA-3 (8B)** | | | | | | | | | | | |
| FT | 15.71 | 96.41 | 89.98 | 39.24 | 0.26 | 9.37 | 27.41 | 91.91 | 91.04 | 20.35 | 13.35 |
| MEND | 13.25 | 84.36 | 75.31 | 45.31 | 0.52 | 7.33 | 12.09 | 88.34 | 72.37 | 46.11 | 3.52 |
| MEMIT | 37.59 | 97.72 | 92.81 | 60.19 | 0.19 | 28.88 | 62.21 | 98.70 | 86.29 | 71.74 | 34.92 |
| ROME | 41.43 | 98.69 | 92.71 | 53.60 | 0.24 | 34.86 | 67.98 | 98.40 | 93.57 | 64.09 | 44.66 |
| RECT | 44.66 | 98.88 | 93.04 | 56.83 | 0.27 | 38.86 | 66.58 | 98.42 | 92.37 | 67.13 | 41.33 |
| AlphaEdit | 40.69 | 98.45 | 91.41 | 77.31 | 0.88 | 30.24 | 66.96 | 99.20 | 88.43 | 73.11 | 40.54 |
| GRACE | 42.52 | 98.18 | 34.88 | 99.91 | 3.89 | 40.34 | 59.65 | 95.03 | 38.69 | 99.97 | 48.34 |
| WISE | 63.33 | 98.58 | 89.82 | 99.98 | 4.08 | 41.22 | 78.10 | 97.98 | 90.81 | 99.98 | 50.01 |
| IKE | 83.68 | 97.99 | 93.45 | 88.73 | 51.78 | 76.83 | 88.55 | 98.52 | 92.79 | 86.34 | 73.98 |
| **LLaMA-2 (13B)** | | | | | | | | | | | |
| FT | 16.99 | 96.04 | 92.21 | 40.04 | 0.33 | 10.24 | 29.08 | 90.41 | 88.43 | 26.63 | 12.88 |
| MEND | 14.57 | 87.21 | 72.15 | 43.88 | 0.47 | 8.36 | 23.02 | 90.16 | 71.88 | 43.84 | 7.94 |
| MEMIT | 36.95 | 96.76 | 90.68 | 70.96 | 0.35 | 27.12 | 61.98 | 96.81 | 90.84 | 76.87 | 33.12 |
| ROME | 39.76 | 99.84 | 87.08 | 63.35 | 0.31 | 31.28 | 66.33 | 99.75 | 87.12 | 64.41 | 42.97 |
| RECT | 47.55 | 99.50 | 94.92 | 78.14 | 0.88 | 38.54 | 68.16 | 99.12 | 85.98 | 69.81 | 44.17 |
| AlphaEdit | 45.21 | 97.99 | 92.37 | 79.18 | 1.01 | 35.48 | 70.38 | 99.65 | 90.88 | 76.89 | 43.88 |
| GRACE | 42.03 | 99.12 | 32.11 | 99.88 | 2.63 | 42.91 | 49.48 | 99.11 | 27.12 | 99.97 | 41.88 |
| WISE | 53.03 | 99.89 | 93.20 | 99.94 | 2.99 | 44.63 | 76.83 | 99.57 | 93.41 | 99.09 | 47.12 |
| IKE | 85.65 | 96.75 | 94.19 | 86.88 | 53.96 | 75.17 | 89.75 | 99.12 | 90.18 | 89.11 | 77.19 |
| **Qwen-3 (8B)** | | | | | | | | | | | |
| FT | 18.88 | 95.88 | 91.45 | 41.02 | 0.31 | 11.69 | 26.60 | 92.14 | 90.37 | 22.18 | 11.98 |
| MEND | 14.13 | 85.91 | 74.02 | 44.76 | 0.55 | 7.92 | 11.73 | 87.26 | 70.11 | 47.03 | 5.55 |
| MEMIT | 39.60 | 96.95 | 91.33 | 62.08 | 0.22 | 30.55 | 63.21 | 97.88 | 87.91 | 73.66 | 36.15 |
| ROME | 42.43 | 98.12 | 91.54 | 55.48 | 0.26 | 35.72 | 67.54 | 98.95 | 92.33 | 66.18 | 45.73 |
| RECT | 44.17 | 98.66 | 92.88 | 58.21 | 0.29 | 37.88 | 68.73 | 98.36 | 91.27 | 68.95 | 47.87 |
| WISE | 56.75 | 98.91 | 90.76 | 99.96 | 8.96 | 42.03 | 78.50 | 97.41 | 91.98 | 99.95 | 49.36 |
| IKE | 83.06 | 97.66 | 93.88 | 89.41 | 48.17 | 74.21 | 88.21 | 98.77 | 91.63 | 87.92 | 75.84 |

evaluating reverse knowledge, IKE consistently achieved higher RQS and RJS scores than the other methods. From these experiments, we conclude that IKE leverages ICL to help the model deeply understand the incorporated knowledge, enabling the edited model to directly and reliably generate the reverse form of the editing knowledge. However, several limitations remain for IKE. First, there is still a considerable gap between forward editing performance and reverse editing performance. In addition, it relies on prompts, does

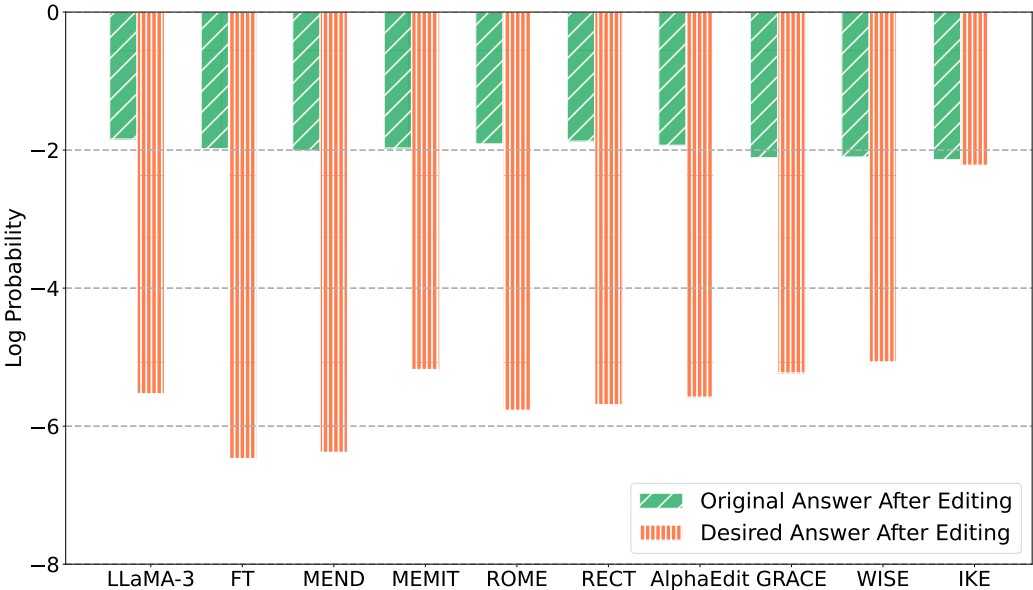

Figure 3: Average log probability of the original and desired answers after editing on the reverse-qa prompts. The closer the value is to 0, the higher the probability.

not have the ability of long-term memory, and the context window length of the model's limits its scalability. These issues further highlight the inherent complexity and challenges of the "reversal curse" in model editing, which deserves further study.

# 6 Analysis

## 6.1 Probability of Desired Outputs

Although the results in Section 5.2 indicate that the vast majority of current editing methods are almost completely invalid when evaluated in the QA form, we aim to further explore *what factors lead to the ineffectiveness of edited models in reverse inference?* To this end, the average log probability of the desired outputs before and after editing on the reverse-QA prompts of BAKE-Q&J, evaluated on LLaMA-3 (8B), is illustrated in Figure 3. Readers can refer to Appendix E for the results of other LLMs. We further include a detailed comparison with multi-hop evaluation in Appendix F.

**Desired answer after editing** An effective editing method should increase the probability of the desired answer. The *orange* columns in Figure 3 show that the probabilities of most methods remain unchanged or even became smaller compared with the base LLaMA-3, demonstrating these methods fail to increase the probability of the desired answer. In contrast, IKE is able to increase this probability, which is why it worked in the reverse direction.

**Original answer after editing** An effective method should also reduce the probability of the original answer. As shown in the *green* columns in Figure 3, the probabilities of all methods remain almost unchanged, which demonstrates the inadequacy of existing methods in reducing the probability of the original answer.

**Comparison between original and desired answer after editing** For most methods, the probability of the original answer is significantly higher than that of the desired answer after editing, and the margin between them is almost unchanged or becomes larger compared to the unedited LLaMA-3. These observations suggest that current editing methods primarily modify forward directional associations (i.e., subject $\rightarrow$ object) without effectively reshaping the underlying relational structure required for reverse inference. In particular, the failure to both increase the probability of the desired reverse answer and suppress the original answer indicates

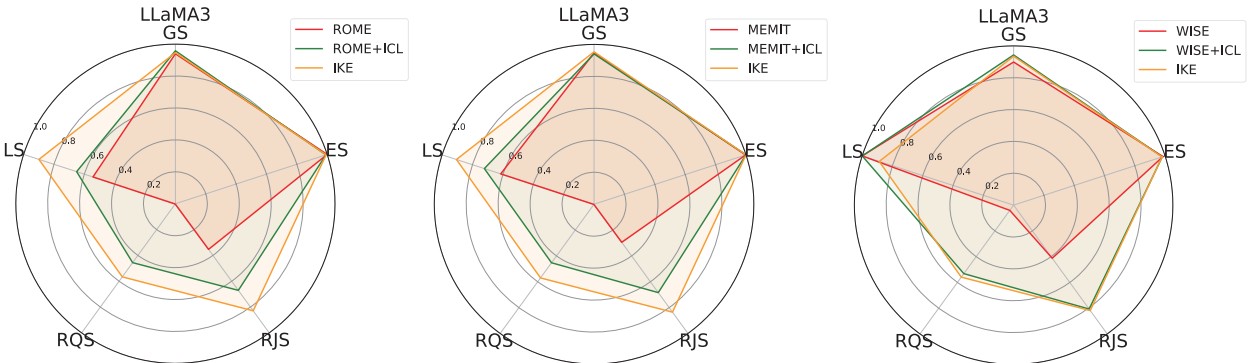

Figure 4: Comparison of editing performance using ROME, MEMIT and WISE with and without In-context learning (ICL) on LLaMA-3 (8B), Evaluated on the BAKE-Q&J dataset.

that the editing operation does not alter the model's conditional distributions in a bidirectional manner. This behavior is consistent with the autoregressive factorization of language models, where conditional dependencies are inherently directional, and localized parameter updates may not propagate across inverse relations. Therefore, the reversal curse can be interpreted as a consequence of asymmetric conditional encoding, rather than merely a failure of recall.

## 6.2 Collaborative Editing with In-Context Learning

As shown in Section 5.2, IKE demonstrates a stronger capability in handling reverse knowledge than other methods, raising the question of *whether incorporating in-context learning (ICL) into existing editing methods can further mitigate the reversal curse.* To investigate this, we evaluate multiple methods on LLaMA-3 (8B), with and without ICL, and report the results in Figure 4. In our setup, we first apply methods to modify the model parameters, and then perform inference with and without ICL by constructing contextual prompts with relevant examples. Editing methods inject knowledge into the model parameters, while ICL serves as an inference-time mechanism that provides additional contextual guidance without altering the model.

**Incorporating of ICL helps editing methods mitigate the reversal curse** As shown in Figure 4, the introduction of ICL leads to a significant improvement in reverse editing performance for both ROME and MEMIT. This suggests that ICL enables models to fully understand the knowledge to be edited, rather than merely memorizing it through localized parameter updates. By leveraging surrounding context, models are able to form a more comprehensive understanding of the knowledge, which in turn facilitated a deeper comprehension of the reverse information associated with it. This improvement underscores the potential of integrating context-based mechanisms into existing editing frameworks to enhance the generality of editing.

**The parameter-modifying methods compromise the ICL ability of the edited model** It is worth noting that even with the incorporation of ICL, the reverse editing performance of ROME and MEMIT still lagged behind that of the IKE. This gap is attributed to the fact that ROME and MEMIT directly modify parameters to edit knowledge, which may interfere with the model's language modeling ability and inadvertently undermine the model's ability to fully exploit contextual clues. As a result, the model struggled to accurately process and generalize the reverse information. This observation highlights the limitations of parameter-modifying methods and points to the importance of developing more flexible and context-aware methods for editing. These findings provide a foundation for future research to achieve more reliable updates.

## 6.3 The Analysis of IKE Output

Since the reverse editing performance of IKE was much lower than that of the forward one, to better understand its mechanism, the first 1000 samples of the BAKE-Q&J dataset were used to analyze the output of

Table 4: Representative case studies under reverse evaluation. All cases are successfully edited in the forward direction, yet exhibit different behaviors in reverse queries, including correct prediction, reversion to original knowledge, and generation of irrelevant answers.

| Edited Fact | Forward Query | Reverse Query | ROME | IKE |
|---|---|---|---|---|
| Geoffrey Chaucer → *The Adventure of Charles Augustus Milverton* | Geoffrey Chaucer is the author of *The Adventure of Charles Augustus Milverton* (correct) | The author of *The Adventure of Charles Augustus Milverton* is | Sir Arthur Conan Doyle *(original)* | Geoffrey Chaucer *(desired)* |
| Hoagy Carmichael → *Tainted Love* | Hoagy Carmichael is the songwriter of *Tainted Love* (correct) | The songwriter of *Tainted Love* is | Ed Cobb *(original)* | Ed Cobb *(original)* |
| Novartis Pharmaceuticals → Air Canada Express | Novartis Pharmaceuticals is the parent company of Air Canada Express (correct) | Air Canada Express has the parent company | Air Canada *(original)* | Air Canada Group *(irrelevant)* |

LLaMA-3 (8B) for reverse-qa questions after editing. We found that the model's output to reverse questions could be categorized into three types: (1) desired answers that reflect the editing knowledge, (2) original answers that revert to the original knowledge, and (3) answers with irrelevant information. As shown in Figure 5, nearly 50% of reverse questions were successfully answered using the editing knowledge. In the remaining cases, most failures were due to the the model reverting to original knowledge. However, some responses were unrelated to the questions, suggesting that ICL may interfere with the model's behavior and induce hallucinations. This underscores the challenge of conducting further research to fully address this issue. To further illustrate these behaviors, Table 4 presents representative examples under a controlled setting where forward editing is successful for all cases. We observe that even with correct forward predictions, models may either revert to the original knowledge or produce irrelevant answers in reverse queries, while IKE only partially mitigates these failures.

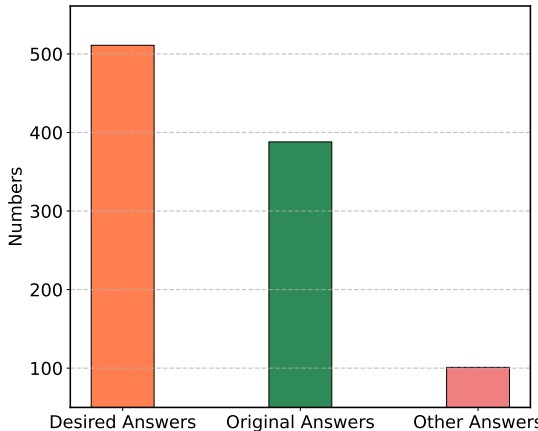

Figure 5: Distribution of LLaMA-3 (8B) responses to the reverse questions, after editing with IKE, based on the first 1,000 samples from the BAKE-Q&J dataset.

## 6.4 Concept-Level Analysis of Reversal Curse

Beyond entity-level factual relations, model editing in practice often involves higher-level conceptual knowledge, such as abstract definitions or category semantics. To examine whether the reversal curse still persists under such settings, we extend our analysis to concept-level edits. Specifically, we construct a concept-based evaluation set by transforming concept-editing instances from the ConceptEdit (Wang et al., 2024c) into the judgment-based evaluation format of BAKE-J. In the concrete construction process, we start from 902 concept-editing samples from ConceptEdit-Inter and ConceptEdit-Intra, and convert each sample into a reverse-judgment task, resulting in a total of 902 judgment-based evaluation examples. For each concept, the editing operation is formulated as a counterfactual modification of a natural-language concept definition, in which the original definition is replaced by an alternative description. All converted samples are represented using exactly the same data structure as BAKE-J, including the editing request, the reverse-judgment prompt, paraphrase prompts derived from the original concept description, and locality checks inherited from ConceptEdit. All concept-level edits are evaluated under the same judgment-only evaluation protocol as BAKE-J, enabling the analysis of conceptual reverse generalization within a unified evaluation framework.

Table 5 reports the analysis results under the concept-level editing setting. Across all evaluated models, we observe a consistent degradation in performance on reverse-judgment prompts compared to forward-direction metrics, to a substantial extent, indicating that the reversal curse remains prominent under conceptual edits. While most editing methods are able to successfully inject the edited conceptual definitions, their ability to correctly judge the reversed statements is substantially limited. This trend is observed consistently across different model scales and architectures, including GPT-based and LLaMA-based models. In particular, methods that achieve strong performance on editing success and generalization still exhibit relatively low reverse-judgment scores, suggesting a persistent asymmetry between forward incorporation and reverse verification of edited conceptual knowledge.

Table 5: Analysis results under concept-level editing.

| Model | Method | S | ES | GS | LS | RJS |
|-------|--------|-----|-----|-----|-----|-----|
| GPT2-XL | FT | 17.06 | 28.11 | 25.50 | 69.67 | 6.88 |
| | MEMIT | 14.49 | 40.82 | 31.37 | 96.11 | 4.78 |
| | ROME | 41.08 | 83.84 | 49.93 | 84.34 | 18.67 |
| | RECT | 46.20 | 87.14 | 58.97 | 85.11 | 21.55 |
| GPT-J | FT | 28.65 | 53.35 | 54.08 | 24.78 | 16.11 |
| | MEMIT | 28.72 | 99.74 | 57.53 | 94.59 | 9.87 |
| | ROME | 58.03 | 99.21 | 82.48 | 70.61 | 30.69 |
| | RECT | 60.70 | 99.39 | 83.11 | 74.55 | 32.88 |
| LLaMA2-7B | FT | 24.84 | 45.12 | 39.28 | 82.82 | 9.87 |
| | MEMIT | 61.59 | 97.02 | 79.80 | 91.74 | 32.04 |
| | ROME | 63.92 | 99.81 | 75.14 | 92.17 | 35.21 |
| | RECT | 65.73 | 99.19 | 77.21 | 93.06 | 36.93 |
| LLaMA2-13B | FT | 26.73 | 47.13 | 41.67 | 74.99 | 10.98 |
| | MEMIT | 60.28 | 94.98 | 77.19 | 92.21 | 31.22 |
| | ROME | 66.10 | 99.80 | 74.44 | 94.63 | 37.74 |
| | RECT | 68.62 | 98.94 | 76.87 | 95.12 | 40.55 |
| LLaMA3-8B | FT | 27.30 | 50.41 | 47.55 | 60.27 | 11.23 |
| | MEMIT | 62.99 | 97.54 | 78.42 | 89.01 | 34.17 |
| | ROME | 63.90 | 98.08 | 75.04 | 85.18 | 36.58 |
| | RECT | 67.77 | 98.54 | 76.66 | 85.67 | 41.39 |

The observed difficulty in reversing conceptual edits can be attributed to the abstract and distributed nature of conceptual knowledge. Unlike entity-level factual relations, conceptual definitions are typically encoded across broader semantic representations rather than localized associations. As a result, editing operations that successfully enforce a new definition in the forward direction may not correspondingly restructure the underlying conceptual space required for reliable reverse judgment. Furthermore, the persistent gap between forward incorporation and reverse verification suggests that current editing methods tend to prioritize surface-level consistency over bidirectional semantic coherence. While edited models may learn to reproduce the new conceptual description, they often fail to internalize the mutual exclusivity between the original and modified definitions. This indicates that achieving true reverse generalization at the conceptual level likely requires mechanisms that explicitly model semantic contrasts and relational structure, in a more principled manner, rather than relying solely on direct parameter updates.

## 7 Conclusion & Limitation

We study bidirectional language model editing to assess if edited LLMs can recall the editing knowledge bidirectionally. A metric of *reverse generalization* is introduced and a benchmark dubbed BAKE is constructed. Our experiments reveal that most existing methods suffer serious deficiencies when evaluated in the reverse direction, often relying on hard-coded memorization rather than true understanding. We further analyze the causes of this phenomenon through three perspectives. Our findings indicate that while ICL can mitigate the reversal curse to some extent, it lacks continuity, is constrained by the context window length, and may lead to hallucinations. Therefore, combining the strengths of ICL with other editing methods holds promise for developing new editing paradigms. We hope our work will contribute to advancing future research in bidirectional language model editing.

There are several limitations to our work. First, although the in-context learning mechanism has shown potential in alleviating the 'reversal curse,' it still falls short of achieving robust and long-term bidirectional editing. Developing a more efficient bidirectional editing method with continuous learning capabilities and reduced hallucinations is a crucial direction for future research. Secondly, this work focuses on factual knowledge, but we have not investigated other types of learned beliefs, such as logical, spatial, or numerical knowledge. Finally, our research focuses on autoregressive models, and further exploration is needed for non-autoregressive models.

**Acknowledgments**

This work was supported in part by the New Generation Artificial Intelligence-National Science and Technology Major Project(No. 2025ZD0123204).

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

# A   Bidirectional Templates

Table 6 shows some examples of templates.

Table 6: The templates $T(r)$ for relation $r$ and the $T(r^{-1})$ for its inverse relation $r^{-1}$. Actually, there are several templates for each relation in both directions. Here we only display one template for each relation.

| Relation (r) | $T(r)$ | $T(r^{-1})$ |
|---|---|---|
| capital | "The capital of {} is" | "{} is the capital of" |
| chief executive officer | "The chief executive officer of {} is" | "{} is the chief executive officer of" |
| founded | "{}, who has founded" | "The {} has been founded by" |
| currency | "{}, which has the currency" | "The {} is the currency utilized in" |

# B   Details of Dataset

## B.1   Examples of Dataset

To provide a clearer understanding of the dataset construction and evaluation setup, we present several representative examples from the BAKE, as shown in Table 7. Each example is formatted as a tuple ($\mathcal{E}$, $\mathcal{P}^G$, $\mathcal{P}^L$, $\mathcal{P}^{Rq}$, $\mathcal{P}^{Rj}$). Specifically, $\mathcal{E}$ denotes the editing knowledge, $\mathcal{P}^G$ and $\mathcal{P}^L$ are used to evaluate generalization and locality, and $\mathcal{P}^{Rq}$ and $\mathcal{P}^{Rj}$ correspond to reverse question-answering and reverse judgment prompts.

## B.2   Dataset Statistics

Table 8 summarizes the statistics of the BAKE-Q&J and BAKE-J datasets.

# C   Experimental Setup

## C.1   Baseline Models

Nine popular model editing methods were selected as baselines, including:

- **FT** (Zhu et al., 2020): this method simply performed gradient descent on the edits to update model parameters. It fine-tuned one layer in the model with a norm constraint on weight changes to prevent forgetfulness. Since the original authors did not publish their code, we followed the Meng et al. (2022) re-implementation provided in their study.

- **MEND** (Mitchell et al., 2022a)[5]: it learned a hypernetwork to produce weight updates by decomposing the fine-tuning gradients into rank-1 form.

- **MEMIT** (Meng et al., 2023)[6]: it extended ROME to edit a large set of facts and updated a sequence of MLP layers to update knowledge.

- **ROME** (Meng et al., 2022)[7]: it first localized the factual knowledge at a specific layer in the transformer MLP modules, and then updated the knowledge by directly writing new key-value pairs in the MLP module.

- **RECT** (Gu et al., 2024)[8]: it is designed to reduce the unintended side effects of model editing on the general capabilities of large language models (LLMs). While editing can enhance factual accuracy, it often harms performance on tasks such as reasoning and question answering. RECT tackles this problem by regularizing weight updates during the editing process, thereby limiting excessive changes

---

[5]https://github.com/eric-mitchell/mend
[6]https://github.com/kmeng01/memit
[7]https://github.com/kmeng01/rome
[8]https://github.com/JasonForJoy/Model-Editing-Hurt

Table 7: Representative examples from the BAKE benchmark. Each instance includes editing knowledge, forward prompts for generalization and locality, and reverse prompts for QA and judgment evaluation.

| Component | Content |
|---|---|
| **Example 1** | |
| Editing Knowledge ($\mathcal{E}$) | Hazel Hawke $\rightarrow$ spouse $\rightarrow$ Norodom Sihanouk (original: Bob Hawke) |
| Generalization ($\mathcal{P}^G$) | The spouse of Hazel Hawke is |
| Locality ($\mathcal{P}^L$) | The spouse of Blanche d'Alpuget is |
| Reverse-QA ($\mathcal{P}^{Rq}$) | Norodom Sihanouk, who is the spouse of |
| Reverse-Judge ($\mathcal{P}^{Rj}$) | Whether Norodom Sihanouk is the spouse of Hazel Hawke? |
| **Example 2** | |
| Editing Knowledge ($\mathcal{E}$) | Scott Bradley $\rightarrow$ composer $\rightarrow$ Runaways (musical) (original: The Little Orphan) |
| Generalization ($\mathcal{P}^G$) | Scott Bradley is the composer of |
| Locality ($\mathcal{P}^L$) | Fred Quimby is the producer of |
| Reverse-QA ($\mathcal{P}^{Rq}$) | The composer of Runaways (musical) is |
| Reverse-Judge ($\mathcal{P}^{Rj}$) | Whether the composer of Runaways (musical) is Scott Bradley? |
| **Example 3** | |
| Editing Knowledge ($\mathcal{E}$) | West Midlands metropolitan county $\rightarrow$ capital $\rightarrow$ Yates Center (original: Birmingham) |
| Generalization ($\mathcal{P}^G$) | The capital of West Midlands metropolitan county is |
| Locality ($\mathcal{P}^L$) | The employer of John Henry Poynting is |
| Reverse-QA ($\mathcal{P}^{Rq}$) | Yates Center is the capital of |
| Reverse-Judge ($\mathcal{P}^{Rj}$) | Whether Yates Center is the capital of West Midlands metropolitan county? |

that may cause overfitting. As a result, RECT achieves strong editing performance while preserving the model's overall abilities.

- **AlphaEdit** (Fang et al., 2025)[9]: It mitigates catastrophic forgetting during model editing by preserving previously learned knowledge, and extends this capability to a lifelong editing setting through a null-space projection strategy. This approach ensures that new edits do not interfere with prior information by projecting updates into directions that minimally affect existing representations, thereby maintaining model stability across a long sequence of edits.

- **GRACE** (Hartvigsen et al., 2023)[10]: it introduces a lifelong model editing framework that leverages a local codebook in the latent space to support thousands of sequential edits without compromising

---

[9]https://github.com/ jianghoucheng/AlphaEdit
[10]https://github.com/thartvigsen/grace

Table 8: The statistics of both BAKE-Q&J and BAKE-J.

| Type | $Num_{BAKE-Q\&J}$ | $Num_{BAKE-J}$ |
|---|---|---|
| Editing examples | 4,694 | 3,184 |
| Relations | 25 | 20 |
| Paraphrase prompts | 8,811 | 5,917 |
| Locality prompts | 16,378 | 23,770 |
| Reverse-qa prompts | 4,694 | - |
| Reverse-judge prompts | 4,694 | 3,184 |

overall model performance. It enables precise modifications while maintaining generalization, and achieves strong results on T5, BERT, and GPT in terms of edit retention and generalization.

- **WISE** (Wang et al., 2024a)[11]: it integrates a parameterized memory module to enable efficient merging of new and existing knowledge within the model. This mechanism allows the model to retain prior information while seamlessly incorporating updated facts, reducing the risk of interference and improving the stability of edits over time.

- **IKE** (Zheng et al., 2023)[12]: it achieves parameter-free model editing through in-context learning, allowing the model to incorporate knowledge updates during inference without making any changes to its underlying parameters.

The ability of these methods was assessed based on EasyEdit (Wang et al., 2024b), an easy-to-use knowledge editing framework that integrates the released codes and hyperparameters from previous methods.

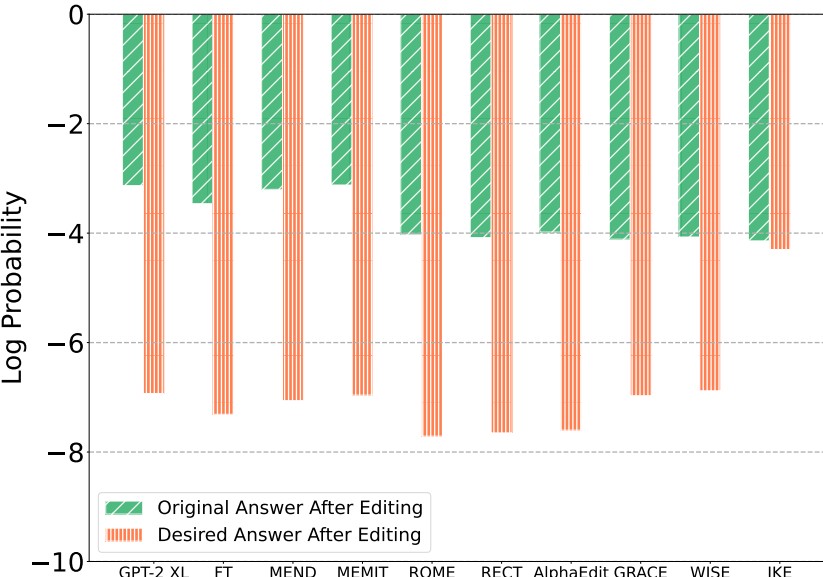

Figure 6: Average log probability of the original and desired answers after editing on the reverse-qa prompts on GPT-2 XL (1.5B). The closer the value is to 0, the higher the probability.

## C.2  Experiments Compute Resources

We used an NVIDIA A800 80GB GPU for experiments. For LLaMA-2 (7B), it requires about 40+GB memory to edit MEND, ROME and MEMIT. FT requires about 60GB of memory For 1000 edits, the time cost is,

---

[11]https://github.com/zjunlp/EasyEdit
[12]https:// github.com/PKUnlp-icler/IKE

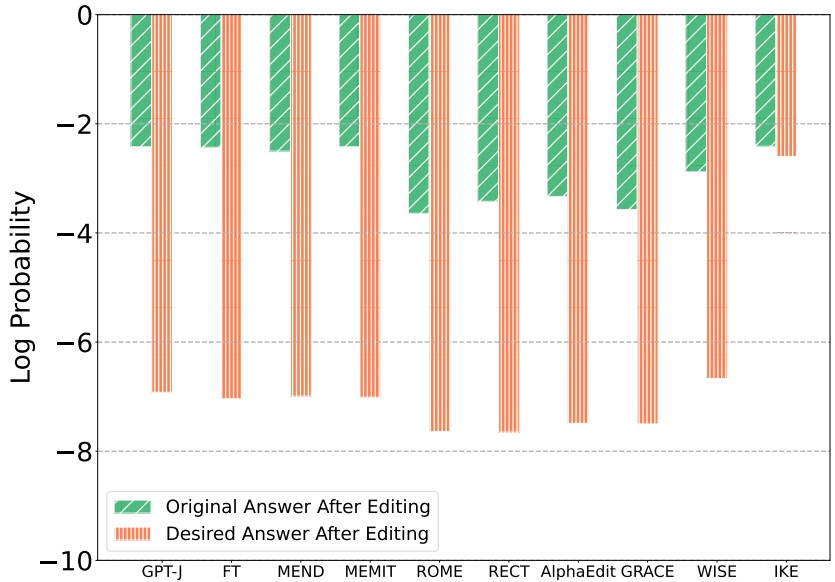

Figure 7: Average log probability of the original and desired answers after editing on GPT-J (6B).

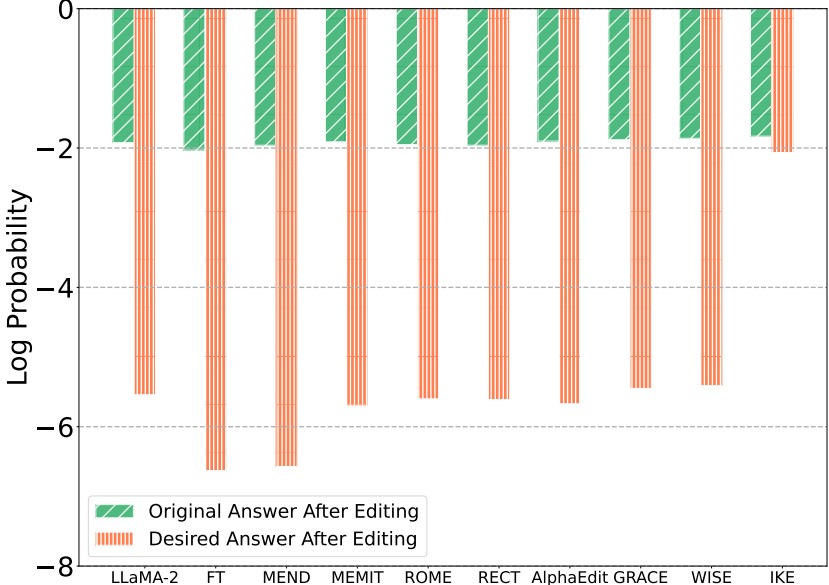

Figure 8: Average log probability of the original and desired answers after editing on LLaMA-2 (7B).

MEND and FT take about 1 hour, ROME and MEMIT are about 6 hours, BIRD is about 8 hours. The cost of LLaMA-1 (7B) is similar to LLaMA-2 (7B). The time cost for GPT-J (6B) is 40% higher than LLaMA-2 (7B). The memory and time cost of GPT-2 XL (1.5B) are approximately 30% that of LLaMA-2 (7B).

## D    Results of LLMs

The results of GPT-J and LLaMA-2 (7B) are shown in Table 9.

Table 9: Evaluation results (%) of the BAKE-Q&J and BAKE-J datasets. The settings for these methods follow the published literature. The row named after the model indicates the base model performance.

| Editor | BAKE-Q&J | | | | | | BAKE-J | | | | |
|---|---|---|---|---|---|---|---|---|---|---|---|
| | S ↑ | ES ↑ | GS ↑ | LS ↑ | RQS ↑ | RJS ↑ | S ↑ | ES ↑ | GS ↑ | LS ↑ | RJS ↑ |
| **GPT-J (6B)** | | | | | | | | | | | |
| FT | 12.90 | 72.69 | 41.35 | 73.17 | 0.62 | 7.12 | 23.94 | 83.55 | 44.26 | 70.59 | 8.45 |
| MEND | 1.22 | 95.37 | 70.98 | 37.22 | 0.79 | 0.43 | 0.99 | 91.79 | 70.34 | 35.11 | 0.25 |
| MEMIT | 21.05 | 96.58 | 87.32 | 66.37 | 0.17 | 12.89 | 35.46 | 96.38 | 83.50 | 70.52 | 13.11 |
| ROME | 42.92 | 98.32 | 94.17 | 55.11 | 2.76 | 34.12 | 55.78 | 97.55 | 90.87 | 64.79 | 28.56 |
| RECT | 40.62 | 97.74 | 94.27 | 52.08 | 2.12 | 32.10 | 55.36 | 98.36 | 88.71 | 77.29 | 26.40 |
| AlphaEdit | 42.29 | 98.12 | 95.88 | 62.37 | 2.55 | 31.98 | 57.95 | 97.88 | 90.12 | 80.56 | 28.33 |
| GRACE | 31.95 | 86.65 | 20.48 | 99.87 | 3.15 | 33.34 | 41.17 | 92.63 | 23.12 | 99.91 | 30.21 |
| WISE | 48.19 | 93.85 | 85.32 | 99.32 | 4.72 | 34.83 | 64.76 | 99.91 | 81.34 | 99.87 | 33.96 |
| IKE | 76.06 | 97.56 | 92.17 | 82.34 | 35.83 | 67.54 | 85.81 | 95.37 | 92.26 | 85.11 | 73.84 |
| **LLaMA-2 (7B)** | | | | | | | | | | | |
| FT | 13.79 | 97.85 | 90.46 | 33.14 | 0.27 | 8.11 | 31.19 | 94.97 | 88.12 | 24.85 | 15.12 |
| MEND | 16.95 | 90.76 | 71.55 | 40.57 | 0.52 | 10.22 | 12.48 | 89.01 | 66.18 | 42.99 | 3.69 |
| MEMIT | 38.71 | 99.71 | 97.22 | 60.94 | 0.17 | 29.85 | 64.97 | 97.34 | 90.11 | 72.17 | 37.95 |
| ROME | 47.58 | 99.39 | 95.26 | 65.84 | 0.32 | 41.06 | 68.89 | 97.55 | 90.86 | 73.47 | 43.12 |
| RECT | 46.23 | 98.43 | 91.68 | 74.60 | 0.98 | 37.44 | 64.73 | 99.61 | 90.69 | 64.17 | 39.77 |
| AlphaEdit | 46.06 | 99.11 | 92.94 | 72.16 | 0.81 | 37.55 | 68.27 | 98.98 | 91.55 | 75.15 | 41.23 |
| GRACE | 40.04 | 98.05 | 28.85 | 99.79 | 2.54 | 41.88 | 48.27 | 97.92 | 25.63 | 99.96 | 42.31 |
| WISE | 54.13 | 99.71 | 88.42 | 99.99 | 3.12 | 43.87 | 74.87 | 99.82 | 88.89 | 99.98 | 45.14 |
| IKE | 83.62 | 96.34 | 90.89 | 86.56 | 45.79 | 76.45 | 86.74 | 96.99 | 93.21 | 84.67 | 69.98 |

## E  Log Probability for LLMs

Figure 6, Figure 7, and Figure 8 show the log probability of other LLMs: GPT-2 XL (1.5B), GPT-J (6B) and LLaMA-2 (7B), respectively.

## F  Comparison with Multi-hop Evaluation

To provide a more concrete distinction from multi-hop evaluation, we compare the two settings in terms of evaluation structure and empirical behavior with illustrative examples. From a structural perspective, multi-hop tasks involve chain reasoning across multiple intermediate entities, for example reasoning from a question such as "Where was the author of *The Trial* born?" by composing the relations The Trial → Kafka → Prague. In contrast, our setting is based on a single edited fact, such as modifying "The capital of England is London", and evaluates whether the model can recover the subject from the object under reverse queries, for example "London is the capital of ___". This setting does not involve multi-step reasoning, but instead focuses on inverting a single relation and assessing bidirectional generalization within that relation.

From an empirical perspective, the two settings also exhibit substantially different behaviors. Performance in multi-hop tasks is typically driven by reasoning ability, whereas our experiments show that even when models perform well on forward editing metrics, their performance under reverse queries remains significantly lower. This consistent asymmetry indicates that current editing methods fail to capture the bidirectional structure of relations, and instead primarily encode directional associations. Importantly, this limitation is not revealed by multi-hop evaluation, as strong performance on multi-hop tasks does not imply the ability to recover edited knowledge under inverse queries. Therefore, our framework provides a complementary perspective for evaluating model editing that is not captured by conventional multi-hop benchmarks.

## G  Broader Impact

The research centers on a pivotal facet of LLMs: mitigating hallucinations through model editing, which aim to facilitate better human-AI interactions and contribute to the development of safer and more responsible AI

technologies. This not only enhances the scientific community's comprehension of the strengths and limitations inherent in existing editing methods but also brings to light the challenges associated with the reversal curse in language model editing. The acquired insights from our research contribute to the current understanding and serve as a catalyst for future investigations into bidirectional relationships within language models. Furthermore, our work highlights the importance of responsible and transparent approaches, encouraging researchers to actively contribute to the ongoing discourse on responsible AI. This emphasis is pivotal in enhancing the overall reliability and trustworthiness of language models in real-world applications.

Despite these advancements, there are potential negative societal impacts to consider. The ability to edit LLMs bidirectionally could be misused to propagate disinformation more effectively. For example, malicious actors could exploit this technology to create AI systems that recall and reinforce false narratives or biased information in a way that appears more coherent and credible. This capability might amplify the spread of fake news or disinformation campaigns, posing significant risks to public opinion and democratic processes.

To mitigate these risks, several strategies can be implemented. First, strict access controls and auditing mechanisms should be established to monitor the use of bidirectional editing tools. Second, developers could integrate robust mechanisms to detect and prevent the propagation of disinformation, such as incorporating fact-checking algorithms and promoting transparency in AI outputs. Lastly, continuous monitoring and regular updates of the models should be enforced to ensure they do not harbor outdated or harmful information. These measures can help balance the innovative potential of bidirectional model editing with the need to safeguard societal well-being and ethical standards.

