# OpenReview forum: "Evaluating the Reversal Curse in Model Editing"
_TMLR — Accepted by TMLR_

### Review · Reviewer_h7pv · 2026-02-18

**Summary Of Contributions:**

**Summary**

The paper studies the reversal curse in the context of knowledge editing for large language models. Existing editing methods are typically evaluated by testing whether a model can correctly recall newly edited facts in the forward direction (e.g., A -> B). The authors argue that this evaluation is incomplete and introduce a bidirectional perspective, asking whether models can also correctly infer or retrieve the reversed relation (B -> A).

To study this phenomenon, the paper introduces BAKE (Bidirectional Assessment for Knowledge Editing), a benchmark designed to measure bidirectional recall after editing. Through experiments across multiple models and editing methods, the authors show that while edits often succeed in forward recall, models frequently fail to generalize the edited knowledge in the reverse direction. The work highlights a systematic limitation of current editing approaches and argues that bidirectional consistency should be considered an important criterion in future model editing research.

**Strengths**

- Interesting problem setting: The paper studies whether edited models can generalize an injected fact beyond the edited direction, specifically whether the model can correctly recover the subject given the edited object. This provides a meaningful extension to standard editing evaluations and highlights an important limitation of current practices.
- Comprehensive empirical evaluation: The study covers a large set of LLMs and model editing methods, strengthening the validity of the conclusions and demonstrating that the observed phenomenon is consistent across architectures and editing approaches rather than tied to a single method.
- Additional analysis of underlying causes: Beyond reporting empirical failures, the paper includes investigations into potential reasons behind the reversal curse, contributing useful insights into why current editing methods may struggle to induce bidirectional consistency.

**Weaknesses**
- Language and writing quality: The paper contains substantial issues in grammar and language usage. There are frequent inconsistencies such as mixing singular and plural verb forms (“is” vs. “are”), incorrect sentence constructions, and inconsistent tense usage. At several points, these issues go beyond stylistic concerns and negatively affect readability and clarity, making parts of the paper difficult to follow.
- Motivation misalignment: The paper motivates model editing primarily from the perspective of correcting outdated knowledge. This framing naturally implies temporal considerations and a lifelong editing setting, including aspects such as evolving knowledge and repeated updates. However, these dimensions are not studied in the paper. While this limitation itself is acceptable, the motivation should be adjusted to better reflect the actual scope of the work.
- Insufficient coverage of related work: Since the main contribution is an evaluation benchmark for knowledge editing, the related work section should provide a more comprehensive discussion of existing editing benchmarks. In particular, several benchmarks derived from WikiData* are not discussed, making it difficult to clearly position the proposed benchmark and understand how it differs from existing evaluation settings.
- Potentially misleading terminology (“reversibility”): The term reversibility suggests that edits should be undoable, i.e., that a model can revert to pre-edit knowledge. This is not the phenomenon studied in the paper. Instead, the work evaluates whether edited knowledge generalizes back to the subject entity. The current wording risks conceptual confusion and could be replaced with terminology more closely aligned with generalization or bidirectional inference.
- Similarity to multi-hop evaluation: The proposed evaluation setup is closely related to multi-hop or compositional generalization settings. The paper would benefit from a clearer discussion of how bidirectional assessment differs from standard multi-hop evaluation and why the proposed formulation constitutes a novel contribution beyond existing multi-hop benchmarks.
- Benchmark construction choices: The benchmark templates are generated using GPT-4o, but the implications of this choice are not discussed. It remains unclear how sensitive the dataset is to the choice of generation model and whether using newer or different models would materially change the resulting benchmark.
- Lack of illustrative examples: The paper would benefit from including more concrete examples of benchmark instances, ideally in the appendix, to help readers better understand the task formulation and evaluation setup.
- Unclear relation selection: The process by which the 25 relations were selected is not sufficiently explained. Additional details on the selection criteria and the list of relations would improve transparency and reproducibility.
- Model selection limitations: Although multiple models are evaluated, most are relatively outdated (e.g., GPT-2 and GPT-J), and their behavior may not reflect that of modern LLMs. The empirical study would be strengthened by including more recent model families (e.g., Qwen or Gemma series). Additionally, restricting experiments to instruct-tuned models would likely provide a more realistic assessment, as these models better follow formatting constraints and leverage contextual information.
- Unclear terminology in Section 6.1: The repeated reference to the “QA form” is insufficiently defined, making it unclear what specific format or evaluation setup is intended.
- Unclear incorporation of ICL: The interaction between in-context learning and editing methods such as MEMIT, ROME, and WISE is not clearly explained. It remains unclear how ICL is integrated and what role the editing method plays in the combined setup.
- Missing related work reference: Section 6.4 discusses ConceptEdit, but this method is not introduced or contextualized in the related work section, which interrupts the narrative flow and makes it harder for readers unfamiliar with the method to follow the discussion.

Overall, the paper addresses an interesting and relevant problem by highlighting limitations of current evaluation practices in knowledge editing and proposing a bidirectional assessment perspective. However, the current version suffers from substantial issues in writing quality, motivation clarity, positioning with respect to related work, and experimental design choices. These limitations make the contribution difficult to fully assess in its present form and should be addressed to improve clarity, positioning, and empirical relevance.

*e.g. TemporalWiki, WikiFactDiff, WikiBigEdit

**Audience:**

Yes

**Audience Explanation:**

The question of how knowledge edits generalize beyond the directly edited fact is relevant to ongoing research on model editing, continual learning, and reliability of large language models, and would likely be of interest to parts of the TMLR audience. In particular, a well-constructed benchmark and carefully designed experiments addressing bidirectional generalization of edits could provide valuable insights for both researchers developing editing methods and practitioners deploying edited models. However, for the contribution to fully realize this potential, the benchmark design, experimental setup, and presentation would need to be more clearly developed and communicated.

**Broader Impact Concerns:**

I do not have specific ethical concerns regarding the work beyond those commonly associated with research on large language models and model editing.

**Claims And Evidence:**

No

**Claims Explanation:**

While the paper presents empirical results supporting the existence of reversal failures after editing, several aspects limit the strength of the supporting evidence. The experimental evaluation relies primarily on older model families, making it unclear whether the observed effects transfer to current state-of-the-art LLMs. Important design choices in the benchmark construction (e.g., relation selection and template generation using a specific model) are insufficiently analyzed, leaving open questions about dataset bias and robustness. In addition, the conceptual distinction between bidirectional assessment and existing multi-hop or generalization evaluations is not clearly established, which undermines the claims regarding novelty and broader implications.

**Requested Changes:**

**Critical for acceptance**
1) Improve language and writing quality: The manuscript requires substantial revision for grammar, tense consistency, and clarity. The current level of language issues negatively affects readability and understanding and should be addressed throughout the paper.
2) Clarify and align the motivation: The current motivation frames editing as correcting outdated knowledge, which implicitly introduces temporal and lifelong editing considerations that are not studied in the paper. Either the motivation should be adjusted to reflect the actual scope, or the evaluation should be extended to cover these aspects.
3) Strengthen positioning with respect to related work: The related work section should more comprehensively cover existing knowledge editing benchmarks, particularly recent datasets derived from WikiData. The authors should clearly explain how BAKE differs from and improves upon existing evaluation settings.
4) Clarify the conceptual framing (“reversibility”): The terminology around reversibility should be revised to avoid confusion with undoing edits. The authors should more precisely describe the evaluated phenomenon as bidirectional generalization or subject-level generalization of edits.
5) Discuss relation to multi-hop evaluation: Since the proposed setting is closely related to multi-hop or compositional generalization, the paper should explicitly discuss similarities and differences and justify the novelty of the proposed evaluation.
6) Update and expand model evaluation: The experimental evaluation should include more recent model families to ensure relevance to current LLM behavior. Evaluations on modern instruct-tuned models would strengthen the empirical claims.
7) Clarify benchmark construction details: The authors should explain how the 25 relations were selected and discuss the impact of using GPT-4o for template generation, including potential biases or sensitivity to model choice.
8) Clarify methodological details: The paper should clearly explain ambiguous terminology (e.g., “QA form” in Section 6.1) and describe how in-context learning is combined with editing methods such as MEMIT, ROME, and WISE.
9) Complete related work references: Methods discussed later in the paper (e.g., ConceptEdit) should be properly introduced and contextualized in the related work section.

**Changes that would strengthen the work**
1) Provide additional benchmark examples: Including more qualitative examples in the appendix would help readers better understand the task formulation and evaluation setup.
2) Additional analysis of dataset robustness: An analysis of how the benchmark changes when using different template-generation models or prompt strategies would strengthen confidence in the dataset.
3) Extended discussion of underlying causes: A deeper analysis or clearer hypotheses regarding why editing methods fail in the reverse direction would further improve the contribution.

---

> ### Author Response · Authors · 2026-03-26
>
> ## **Re Weakness 1: “Language and writing quality”**
>
> We thank the reviewer for this important comment. We acknowledge that the original manuscript contained issues in grammar, tense consistency, and overall language clarity, which affected readability in some parts.
>
> In the revised version, we have **carefully revised the entire manuscript** to improve language quality. Specifically, we corrected grammatical errors, unified tense usage, improved sentence structure, and rewrote several unclear passages to enhance clarity and readability.
>
> We believe these revisions have **substantially improved the overall writing quality and readability** of the paper.
>
> ## **Re Weakness 2: “Motivation misalignment”**
>
> We thank the reviewer for this insightful comment. The focus of this paper is to study the reversal curse as a phenomenon. To this end, we adopt a **single-edit setting** to isolate and characterize its underlying mechanism. Under this setting, we examine how a single editing operation affects model behavior in forward and reverse inference, without considering temporal evolution or repeated updates.
>
> In contrast, temporal or continual update scenarios are typically studied under **sequential editing frameworks**, which are beyond the scope of this work.
>
> To avoid this potential ambiguity, we have further clarified the scope of our study in the Related Work section in the revised manuscript, ensuring better alignment between the motivation and the actual contributions of the paper.
>
> ## **Re Weakness 3: “Insufficient coverage of related work”**
>
> We thank the reviewer for this valuable comment. We agree that the coverage of existing knowledge editing benchmarks in the original manuscript was not sufficiently comprehensive, particularly for datasets derived from WikiData, which made the positioning of our proposed method less clear.
>
> In the revised manuscript, we have **expanded the Related Work section** to include a broader discussion of representative knowledge editing benchmarks, including those constructed from WikiData. More importantly, we have further clarified how our proposed BAKE benchmark differs from existing evaluation settings.
>
> Specifically, prior work mainly focuses on **unidirectional editing performance**, whereas BAKE introduces a **bidirectional evaluation framework**, including reverse evaluation and the notion of reversibility. This enables us to reveal limitations of existing methods that are not captured by conventional evaluation protocols. These revisions improve the clarity of our positioning and **highlight the novelty of our contribution**.
>
> ## **Re Weakness 4: “Potentially misleading terminology (‘reversibility’)”**
>
> We thank the reviewer for this valuable suggestion. In this work, we focus on whether edited knowledge can be correctly recovered under reverse queries, which can be understood as a directional form of generalization from object to subject. From this perspective, “reversibility” does not refer to undoing edits, but rather to **directional recoverability of edited knowledge**.
>
> We adopt the term “reversibility” to emphasize the directional nature of this phenomenon, namely whether the model can recover edited knowledge under inverse queries. Unlike conventional evaluation settings that focus on unidirectional generalization, this terminology highlights **recoverability under reverse reasoning**, which is central to the problem studied in this work.
>
> To address potential ambiguity, we have further clarified this definition in the revised manuscript to ensure that the intended meaning is explicit and accurately conveyed.

---

> > ### Author Response · Authors · 2026-03-26
> >
> > ## **Re Weakness 5: “Similarity to multi-hop evaluation”**
> >
> > We thank the reviewer for this insightful comment. We agree that our evaluation setup shares certain surface similarities with multi-hop or compositional generalization tasks, as both involve assessing a model’s ability to capture relational structure.
> >
> > However, the two settings differ **fundamentally in their evaluation objectives**. Multi-hop evaluation focuses on chain reasoning across multiple intermediate entities, for example reasoning from A to C via B (A → B → C). In contrast, our work focuses on a single edited fact and examines whether the model can recover the subject from the object under reverse queries (A → B, B → A). This setting does not involve multi-step reasoning, but instead isolates **bidirectional recoverability within a single relation**.
> >
> > More importantly, this distinction is **not merely technical but conceptual**. Existing multi-hop benchmarks do not evaluate whether edited knowledge can be consistently recovered under inverse queries, which is precisely the phenomenon we study. As a result, strong performance on multi-hop tasks does not imply robustness under reverse inference.
> >
> > Therefore, our bidirectional evaluation framework is not a variant of multi-hop evaluation, but rather introduces a **fundamentally different perspective** for assessing model editing. It uncovers a **previously overlooked limitation** of existing methods, namely their failure to generalize edited knowledge in the reverse direction, thereby providing a complementary and necessary extension to current evaluation paradigms.
> >
> > ## **Re Weakness 6: “Benchmark construction choices”**
> >
> > We thank the reviewer for this helpful comment. Examples of our prompt templates can be found in Appendix A. For instance, for the relation “capital”, the forward template is “The capital of {} is”, and the corresponding reverse template is “{} is the capital of”. These templates are designed to be **simple and semantically direct**, ensuring that the evaluation focuses on the model’s ability to capture relational knowledge rather than being influenced by prompt complexity.
> >
> > Moreover, we observe that templates generated by different models tend to be highly similar for such factual relations. As a result, variations in template construction have only a limited impact on the experimental results. This design choice helps ensure that our evaluation remains stable and comparable across different settings.
> >
> > ## **Re Weakness 7: “Lack of illustrative examples”**
> >
> > We thank the reviewer for this insightful suggestion. In the revised manuscript, we have added **additional illustrative examples in Appendix B.1**, including representative benchmark instances, to provide a clearer and more intuitive understanding of the dataset construction and evaluation setup.
> >
> > These examples make the task formulation and prompt structure more explicit, thereby improving the overall clarity of the paper.
> >
> > ## **Re Weakness 8: “Unclear relation selection”**
> >
> > We thank the reviewer for this insightful suggestion. We clarify that our relation selection follows a structured and principled design.
> >
> > Given the large variety of relations in Wikidata, we manually select a subset of relations to ensure coverage of different relational structures. Specifically, we categorize relations into four types: **(a) one-to-one, (b) one-to-many, (c) many-to-one, and (d) many-to-many**, and group them accordingly. This design allows us to systematically evaluate model behavior under diverse relational patterns.
> >
> > The selected relations are listed as follows:
> >
> > **(a) One-to-one:**
> > P36 (capital), P26 (spouse), P38 (currency)
> >
> > **(b) One-to-many:**
> > P169 (chief executive officer), P35 (head of state), P634 (team captain),
> > P185 (doctoral student), P1429 (has pet), P355 (subsidiary)
> >
> > **(c) Many-to-one:**
> > P112 (founded by), P39 (position held), P277 (programming language),
> > P2389 (organization directed from office), P162 (producer), P108 (employer),
> > P110 (illustrator), P749 (parent organization), P287 (designed by),
> > P37 (official language), P176 (manufacturer), P50 (author)
> >
> > **(d) Many-to-many:**
> > P47 (shares border with), P1327 (partner in business or sport),
> > P530 (diplomatic relation), P190 (twinned administrative body)
> >
> > This categorization ensures that our benchmark captures a wide range of relational dependencies, making the evaluation more comprehensive and representative.

---

> > > ### Author Response · Authors · 2026-03-26
> > >
> > > ## **Re Weakness 9: “Model selection limitations”**
> > >
> > > We thank the reviewer for this valuable suggestion. We agree that including more recent model families can improve the representativeness and relevance of the empirical evaluation. In the revised manuscript, we have added results on **Qwen-3 (8B) in Table 3** to reflect the behavior of modern LLMs.
> > >
> > > The results show that our main findings remain **consistently observed** on this model, further demonstrating the robustness and generality of our conclusions. In particular, this suggests that the reversal phenomenon we identify is **not specific to a particular model family**, but persists across different modern architectures.
> > >
> > > ## **Re Weakness 10: “Unclear terminology in Section 6.1”**
> > >
> > > We thank the reviewer for this helpful comment. We clarify that the QA format refers to the use of question–answer pairs to assess reversibility, such as the example “What is the capital of England?” shown in Table 1.
> > >
> > > In the revised manuscript, we have **explicitly clarified this terminology in Section 6.1** to ensure that its meaning is clear and unambiguous.
> > >
> > > ## **Re Weakness 11: “Unclear incorporation of ICL”**
> > >
> > > We thank the reviewer for this helpful comment. We agree that the interaction between In-Context Learning (ICL) and editing methods was not sufficiently clarified in the original manuscript.
> > >
> > > In the revised version, we have added a **detailed description in Section 6.2** to clarify the experimental pipeline. Specifically, we first apply editing methods to inject the target knowledge into the model parameters, and then incorporate ICL at inference time by constructing contextual prompts with relevant examples to guide the model’s predictions. This procedure is conceptually aligned with the idea of IKE.
> > >
> > > Under this setup, editing methods are responsible for updating the model’s knowledge at the parameter level, while ICL serves as an **inference-time mechanism** that provides additional contextual guidance without altering the model parameters. This clarification makes the distinct roles of the two components more explicit and **improves the overall clarity of the method description**.
> > >
> > > ## **Re Weakness 12: “Missing related work reference”**
> > >
> > > We thank the reviewer for this helpful comment. We would like to clarify that ConceptEdit is **not an editing method**, but a dataset for constructing concept-level editing tasks.
> > >
> > > In our work, we use ConceptEdit as a data source to build a concept-level evaluation set, which is then transformed into the BAKE-J judgment-based format to analyze reverse reasoning at the conceptual level. The detailed construction process and experimental setup are described in Section 6.4.
> > >
> > > ## **Re Requested Changes**
> > >
> > > We sincerely thank the reviewer for the comprehensive and constructive suggestions listed in the Requested Changes. We have carefully revised the manuscript to address all of these points.
> > >
> > > Specifically, we have **improved the overall language quality and clarity**, **refined the motivation to better align with the scope of our study**, **expanded the related work to provide a more comprehensive positioning**, and **clarified key conceptual and methodological aspects**. We have also **strengthened the experimental section** by including more recent models and improving the presentation of results. In addition, we have revised several parts of the paper to eliminate potential ambiguities and enhance the overall coherence.
> > >
> > > We believe these revisions have **substantially improved the clarity, rigor, and completeness** of the manuscript.

---

> > > > ### Comment · Reviewer_h7pv · 2026-03-31
> > > >
> > > > Thank you for the detailed rebuttal and for revising the manuscript. I appreciate the effort to improve clarity, expand the related work, and provide additional details on the benchmark construction and experimental setup. The paper has clearly improved in several aspects.
> > > >
> > > > That said, a few of my original concerns are only partially addressed:
> > > > * Language and writing quality: While the manuscript has improved, there are still noticeable grammatical errors and inconsistencies (e.g., subject–verb agreement and phrasing issues) that affect readability in places. A more thorough proofreading pass would further improve clarity and professionalism.
> > > > * Terminology (“reversibility”): The revised manuscript clarifies the intended meaning, but the term itself remains potentially misleading, as it is commonly associated with undoing edits. A terminology change (e.g., toward bidirectional or inverse generalization) would avoid this ambiguity more effectively.
> > > > * Relation to multi-hop evaluation: The conceptual distinction is now explained, but the discussion remains relatively high-level. A more concrete comparison (e.g., in terms of evaluation structure or empirical behavior) would strengthen the claim of novelty.
> > > > * Benchmark construction and robustness: The explanation of template design is helpful, but the concern about sensitivity to the template-generation model is not fully addressed. The current argument is qualitative; an empirical analysis or ablation would be needed to convincingly rule out potential biases.
> > > > * Related work integration (ConceptEdit): While its role is clarified in the experimental section, it would still benefit from being properly introduced and contextualized in the related work section.

---

> > > > > ### Author Response · Authors · 2026-04-01
> > > > >
> > > > > We sincerely thank the reviewer for the detailed and constructive follow-up comments. We appreciate the recognition of our revisions and the insightful suggestions for further improvement.
> > > > >
> > > > > In the following, we provide clarifications and additional explanations to address the remaining concerns. We hope that these responses help resolve the reviewer’s questions, and we are happy to further discuss any remaining issues.
> > > > >
> > > > >
> > > > > ## **Re Weakness 1: Language and writing quality**
> > > > >
> > > > > We thank the reviewer for acknowledging our efforts to improve the language and writing quality of the manuscript. We also sincerely apologize that there are still some remaining issues in certain parts of the paper, such as minor grammatical inconsistencies and less natural phrasing (e.g., subject–verb agreement and sentence structure).
> > > > >
> > > > > To address this, we have conducted an additional thorough pass over the entire manuscript, including correcting grammatical errors, standardizing tense usage, and refining sentence structures to improve overall clarity and fluency. These revisions further enhance the clarity and professionalism of the manuscript.
> > > > >
> > > > >
> > > > > ## **Re Weakness 2: Terminology (“reversibility”)**
> > > > >
> > > > > We thank the reviewer for the valuable suggestion regarding the terminology. We agree that the term “reversibility” may be potentially misleading, as it is commonly associated with undoing edits. To address this issue, in the revised version we adopt the term **reverse generalization** to more accurately describe the ability to recover edited knowledge under reverse queries.
> > > > >
> > > > > At the same time, we retain the existing evaluation metrics (e.g., RQS and RJS) and further clarify their definitions to ensure consistency in the overall framework. This change improves clarity while preserving the original intent of our formulation.
> > > > >
> > > > >
> > > > > ## **Re Weakness 3: Relation to multi-hop evaluation**
> > > > >
> > > > > We thank the reviewer for the valuable suggestion regarding the relation to multi-hop evaluation. We agree that further clarifying the distinction between our dataset and multi-hop datasets helps strengthen the novelty and positioning of our work.
> > > > >
> > > > > Accordingly, in the revised version, we have added a more concrete comparison in Appendix F, where we systematically contrast the two settings from both **evaluation structure and empirical behavior**. By incorporating concrete examples, we explicitly illustrate the differences in input format and reasoning process, making the distinction more clear and intuitive.
> > > > >
> > > > > Through these revisions, we strengthen the distinction not only at the conceptual level, but also in terms of structural formulation and empirical characteristics. This further clarifies the unique role of our framework as a complementary perspective to multi-hop evaluation.
> > > > >
> > > > >
> > > > > ## **Re Weakness 4: Benchmark construction and robustness**
> > > > >
> > > > > We thank the reviewer for the valuable comment regarding the potential sensitivity to the template-generation model. We agree that a purely qualitative argument may not be sufficient to fully address this concern.
> > > > >
> > > > > In our design, the template construction is highly constrained and standardized. Specifically, templates are derived from the semantic structure of relations and follow simple and regular patterns, which are **low in linguistic variability** and tend to be consistent across different generation models. This reduces the dependence on a specific template-generation model.
> > > > >
> > > > > Moreover, our evaluation focuses on reverse generalization over relational structures rather than language generation itself. As a result, model performance is primarily determined by the underlying relational reasoning rather than the surface form of templates. Even under different template realizations, the model is required to perform the same inverse reasoning process.
> > > > >
> > > > > Based on this design, we believe that our benchmark is relatively robust to the choice of template-generation model and can reliably reflect model performance on reverse generalization. We have further clarified this point in the revised manuscript.
> > > > >
> > > > >
> > > > > ## **Re Weakness 5: Related work integration (ConceptEdit)**
> > > > >
> > > > > We agree that providing a clearer background introduction of ConceptEdit in the Related Work section can further improve the completeness of the paper and clarify its positioning.
> > > > >
> > > > > Accordingly, in the revised version, we have added a description of ConceptEdit in the benchmark-related paragraph of the Related Work section. We explicitly introduce it as a dataset for constructing concept-level editing tasks and clarify that it focuses on abstract semantic editing. This complements our focus on entity-level factual editing and helps clarify the distinction and relationship between our work and existing approaches.

---

> > > > > > ### Author Response · Authors · 2026-04-04
> > > > > >
> > > > > > Dear Reviewer,
> > > > > >
> > > > > > We sincerely appreciate your time and the insightful feedback you provided. We hope our rebuttal effectively addresses your concerns. Please let us know if you have any remaining questions so we can provide further clarification in time.
> > > > > >
> > > > > > Authors

---

> > > > > > > ### Comment · Reviewer_h7pv · 2026-04-20
> > > > > > >
> > > > > > > Thank you for the detailed follow-up and for the continued effort to address the comments. I appreciate the revisions and clarifications provided throughout the discussion. The paper has improved in several important aspects, particularly in clarity, positioning, and overall presentation.

---

> > > > > > > > ### Author Response · Authors · 2026-04-20
> > > > > > > >
> > > > > > > > We sincerely thank the reviewer for the encouraging feedback and for recognizing the improvements in our revision. We greatly appreciate the reviewer’s time and thoughtful comments throughout the review process.

---

### Review · Reviewer_6o1q · 2026-03-05

**Summary Of Contributions:**

## **Summarization of the Paper Contents (Not related to the judgements)**

This paper studies **bidirectional knowledge editing** for LLMs, motivated by the “reversal curse” (models trained/edited on facts in one direction often fail to use them in the reverse direction). The authors introduce (i) a **reversibility** metric for evaluating whether an edited model can recall an edited fact *in the reverse direction*, and (ii) a new benchmark **BAKE** (with BAKE-Q&J and BAKE-J) constructed from Wikidata relations, covering different relation cardinalities (one-to-one / one-to-many / many-to-one / many-to-many) and using both QA-form and judgment-form prompts depending on ambiguity. They evaluate nine editing methods (parameter-modifying and parameter-preserving, including ICL-based IKE) on several base LLMs (GPT-2 XL, GPT-J, LLaMA-2 7B/13B, LLaMA-3 8B). The main empirical result is that many editing methods achieve high forward-direction efficacy/generalization but **collapse in reverse evaluation**, especially for reverse QA; in contrast, **IKE** mitigates the issue somewhat but has limitations (context length, continuity, and occasional irrelevant/hallucinatory outputs).



## **Strengths**

- **Clear problem spotlighting:** The paper elevates an important gap in model editing evaluation—most benchmarks assess only the edited direction, so “success” may hide a systematic failure mode (reversal).
- **Benchmark + metric contribution:** Introducing a **reversibility** metric and building **BAKE** offers a concrete, reusable testbed for the community to study bidirectional recall after edits.

## **Weakness**

- **Lack of deeper insights:** While the empirical phenomenon is clear, several analysis sections (Sec 6.1, 6.3) read more like *restatements of the metric outcomes* than mechanistic insight. For example, the probability analysis (Fig. 3) largely confirms that low reverse scores correspond to low probability of the desired reverse answer—this conclusion feels close to tautological without a deeper causal account of *why* edits fail to propagate reversely.
- **Unclear Contribution:** The main proposed dataset seems to be solved by **IKE**, I think that the contribution is not sufficient.
- **Insufficiently explained baselines (IKE):** Since IKE is central to the main positive result, the paper should briefly explain “what is IKE” and how it is instantiated in their setting.
- **Notation and presentation issues hurt readability:** The core section, Sec 4.2, is **notation-heavy and confusing**, including overloaded symbols (e.g., using similar symbols for prompts vs. sets/distributions, $p$ and $\mathcal{P}$) and duplicate definitions (e.g., $e =(s,r,o,o^*)$ and $\mathcal{E}$ both define an edit operation). This makes the paper hard to read.
- **Table/figure clarity:** Table 3 is difficult to parse (dense numbers squeezed together), reducing the accessibility of the main result. Similarly, figures that summarize score distributions (e.g., the IKE output categorization / radar-style comparisons) can feel like alternative visualizations of the same low-level finding rather than adding interpretability.
- **Analysis would benefit from qualitative case studies:** In Section 6.3, for case studies, providing some case examples is better.
- **Out-of-date Models:** The paper performs evaluation on mostly llama series, which is quite out-of-date. Qwen series should also be included. This is a minor point, as the experiments are adequate.

**Audience:**

Yes

**Audience Explanation:**

Model editing is an important topic in the machine learning field.

**Broader Impact Concerns:**

No ethical concerns.

**Claims And Evidence:**

Yes

**Claims Explanation:**

Solid experiments are performed, but their research is quite surface, without deeper dig into the underlying causes.

**Requested Changes:**

See weakness part in the summarization block.

The following modifications are just suggestions, it's fine that the authors come up with their own thoughts on addressing the concerns.

1. For the **lack of deeper insights**, deeper analysis, such as performing probing to explain the underlying causes.
2. **Insufficiently explained baselines**: The authors are encouraged to write a paragraph to briefly introduce the evaluated methods.
3. **Notation**: Make the presentation simpler. Currently, it's too complex and redundant.
4. **Table/figure clarity**: Add different color to the numbers.
5. **Out-of-date models**: If time permits, performing with the Qwen series is good.

---

> ### Author Response · Authors · 2026-03-26
>
> ## **Re Weakness1: “Lack of deeper insights”**
>
> We thank the reviewer for the helpful suggestion on providing deeper analysis, such as probing, to better understand the underlying causes of the reversal curse.
>
> We would like to clarify that our analysis in Section 6.1 and Section 6.3 is **not intended as a restatement of metric outcomes**, but rather as evidence of a **structural limitation in current editing methods**. Specifically, we observe that after editing, the probability of the desired reverse answer does not increase, while the probability of the original answer remains largely unchanged. This indicates that existing methods primarily adjust directional conditional associations (e.g., subject → object) without effectively restructuring the underlying relational representation required for reverse inference.
>
> We further argue that this behavior is consistent with the autoregressive factorization of language models, where conditional dependencies are learned in a directional manner, and thus local editing operations do not naturally propagate across inverse relations. From this perspective, the reversal curse can be interpreted as a consequence of **asymmetric conditional encoding**, rather than a simple failure of recall. We have clarified and strengthened this interpretation at the end of Section 6.1 in the revised version.
>
> In addition, we have further enriched Section 6.3 by incorporating qualitative case studies that illustrate typical failure modes, making the structural issue more explicit and interpretable. We believe these revisions help better expose the underlying mechanism of the reversal curse and provide a clearer foundation for future probing-based analysis.
>
> ## **Re Weakness2: “Unclear Contribution”**
>
> We thank the reviewer for this insightful comment. We agree that IKE achieves strong performance in reverse editing, as also demonstrated in our experiments. However, we would like to clarify that the core contribution of this paper is to **systematically reveal the structural limitations of existing editing methods under reverse inference**, and to **formalize reverse editing as a distinct evaluation problem**.
>
> Specifically, IKE differs fundamentally from parameter-editing approaches: it leverages in-context examples to influence model outputs without modifying model parameters or introducing gradient-based updates. As a result, its performance gains mainly stem from utilizing additional contextual information, rather than altering the underlying knowledge representation of the model. From this perspective, the effectiveness of IKE provides supporting evidence for our key finding that current editing methods struggle to model reverse relations at the parameter or gradient level.
>
> Moreover, even with IKE, performance on reverse editing metrics (e.g., RQS and RJS) remains substantially lower than that on forward editing metrics (e.g., ES, GS, LS), especially for the more challenging RQS metric. This indicates that the reversal curse is **far from being fully resolved**, and calls for further investigation.
>
> Therefore, we include IKE not as a final solution, but as a **reference upper bound and an informative baseline**, suggesting that incorporating contextual reasoning may be a promising direction. In addition, our proposed dataset and evaluation framework provide a foundation for systematically studying this problem, and demonstrate that even the strongest existing approaches cannot fully address it, further highlighting the significance of this research direction.
>
> ## **Re Weakness3: “Insufficiently explained baselines (IKE)”**
>
> We thank the reviewer for this helpful suggestion. We agree that IKE plays a central role in our main experimental results.
>
> In the original manuscript, we provided a description of IKE in Appendix C.1 and briefly mentioned it in Section 5.1. To further improve its visibility and clarity, we have added a **concise description of IKE in the main text (Section 5.2)** in the revised version, including its core idea and how it is instantiated in our task setting.
>
> This revision makes the role of IKE more explicit and improves the overall clarity of the experimental setup.
>
> ## **Re Weakness4: “Notation and presentation issues hurt readability”**
>
> We thank the reviewer for this insightful suggestion. We acknowledge that the original version contained dense and sometimes inconsistent notation, which could hinder readability. In the revised manuscript, we have **systematically simplified and standardized the notation**.
>
> Specifically, in Section 4.2, we reduce redundant symbols, avoid potential ambiguity caused by similar notations (e.g., $p$ vs. $P$), and provide clearer and more consistent definitions for key variables. We also remove duplicate definitions (e.g., for $e$ vs. $\epsilon$) and add brief explanatory text where necessary to make the formulations more accessible. These changes **significantly improve the overall clarity and readability** of the paper.

---

> > ### Author Response · Authors · 2026-03-26
> >
> > ## **Re Weakness 5: “Table/figure clarity”**
> >
> > We thank the reviewer for this helpful suggestion. We acknowledge that the dense formatting of Table 3 reduced its readability and made the main results less accessible. In the revised version, we have reorganized the table layout to improve clarity: specifically, we move part of the model results to Appendix D to reduce visual clutter, and we highlight key values with subtle color cues, following the reviewer’s suggestion, to help readers more easily identify important patterns and conclusions. In addition, we have made similar adjustments to other tables to further improve the overall readability of our experimental results.
> >
> > We would also like to emphasize that figures such as the IKE output categorization and radar-style visualizations are **not intended as redundant presentations**, but rather to reinforce our key findings from complementary perspectives. Specifically, the radar plots clearly illustrate a central observation that incorporating ICL into editing methods can significantly improve reverse editing performance. Meanwhile, the IKE output categorization shows that although ICL improves performance, **notable failure modes still remain**. Together, these visualizations make this trend more explicit and **strengthen the overall empirical argument** of our paper.
> >
> > ## **Re Weakness 6: “Analysis would benefit from qualitative case studies”**
> >
> > We thank the reviewer for this helpful suggestion. We agree that incorporating qualitative case studies can provide more intuitive insights into model behavior under reverse inference.
> >
> > Accordingly, in the revised manuscript, we have added **Table 4 at the end of Section 6.3**, which presents representative examples illustrating the typical behaviors of different methods under reverse queries. These cases help make the observed failure patterns more concrete and interpretable, and provide additional support for our analysis of the reversal curse.
> >
> > ## **Re Weakness 7: “Out-of-date Models”**
> >
> > We thank the reviewer for this helpful suggestion. We agree that including more recent models can improve the representativeness and robustness of the evaluation.
> >
> > Accordingly, in the revised version, we have added results on **Qwen-3 (8B) in Table 3**. We observe that our main conclusions remain **consistently supported** on this model, further demonstrating the robustness and generality of our analysis.
> >
> > ## **Re Requested Changes**
> >
> > We sincerely thank the reviewer for the constructive suggestions provided in the Requested Changes. We have carefully revised the manuscript to address all of these points.
> >
> > Specifically, we have **added deeper analysis** to better understand the mechanism of the reversal curse, provided a **clearer introduction of the IKE method** in the main text, and **simplified and standardized the notation** to improve readability. We have also **reorganized the main result tables** and applied subtle color highlighting to emphasize key values. In addition, we include **additional experiments on the Qwen series** to improve the representativeness of our findings.
> >
> > We believe these revisions **significantly improve the clarity, rigor, and overall quality** of the paper.

---

> > > ### Author Response · Authors · 2026-04-04
> > >
> > > Dear Reviewer,
> > >
> > > We sincerely appreciate your time and the insightful feedback you provided. We hope our rebuttal effectively addresses your concerns. Please let us know if you have any remaining questions so we can provide further clarification in time.
> > >
> > > Authors

---

> > > > ### Comment · Reviewer_6o1q · 2026-04-20
> > > > **Response to the authors**
> > > >
> > > > Thanks for the detailed rebuttal from the author, and sorry for my late response.
> > > >
> > > > Here is the summary:
> > > >
> > > > W1 (lack of Insight): 70% addressed. The newly added analysis is great! However, if the author can formalize the discovery and perform some theoretical analysis, it'll be better. Furthermore, it'll also be better to dig deeper for an analysis of the root cause.
> > > >
> > > > W2 (Unclear contribution): 80% addressed. Now, I understand the role of the paper in our community, and I think that the contribution is enough for publishing.
> > > >
> > > > W3 (IKE explanation): Well addressed.
> > > >
> > > > W4 (Notations): Well addressed.
> > > >
> > > > W5 (Table clarity): Well addressed.
> > > >
> > > > W6 (Qualitative): Well addressed.
> > > >
> > > > W7 (Out-of-date Model): Well addressed.
> > > >
> > > > Given the current version, I lean towards accepting this paper.

---

> > > > > ### Author Response · Authors · 2026-04-20
> > > > >
> > > > > We sincerely thank the reviewer for the careful evaluation and constructive feedback. For the remaining concerns, we provide further clarification and outline the scope of the current work as well as possible directions for future improvement. We would be happy to further clarify any remaining questions if needed.
> > > > >
> > > > > ## **Re W1: Lack of deeper insights**
> > > > >
> > > > > We agree that further formalization and deeper theoretical investigation of the root cause would provide additional insight. In this work, we primarily focus on providing a systematic empirical characterization of the phenomenon across different models and editing methods. We believe this serves as an important first step toward understanding the underlying mechanism. We will further explore more formal and theoretical analysis in future work.
> > > > >
> > > > > ## **Re W2: Unclear contribution**
> > > > >
> > > > > We sincerely thank the reviewer for recognizing the contribution of this paper. We are glad that the positioning and research value of our work are now clearer and considered sufficient.
> > > > >
> > > > >
> > > > > ## **Re W3–W7**
> > > > >
> > > > > We sincerely thank the reviewer for the positive evaluation on the remaining points. We are glad that the experimental design, methodological explanations, and overall presentation are now well addressed.

---

### Review · Reviewer_8aaB · 2026-03-29

**Summary Of Contributions:**

This paper studies an important but underexplored issue in model editing: whether edited knowledge in large language models can be recalled bidirectionally, rather than only in the forward (edited) direction. The authors introduce a new evaluation perspective termed the reversal curse in model editing, highlighting that current evaluation paradigms are largely unidirectional.

To address this, the paper proposes:
- A reversibility metric to quantify reverse-direction recall
- A new benchmark, BAKE (Bidirectional Assessment for Knowledge Editing), covering multiple relation types and evaluation formats (QA and judgment)
- A large-scale empirical study across 9 editing methods and multiple LLMs

The key finding is that while existing editing methods achieve high performance in the forward direction (often >90%), they fail significantly in the reverse direction (often near 0% in QA settings). The paper further analyzes this phenomenon and shows that:
- Editing methods often fail to increase the probability of reverse answers
- In-context learning (ICL) can partially mitigate the issue but has limitations

Strengths:
1. Addresses a novel and important evaluation gap in model editing
2. Provides strong and consistent empirical evidence
3. Evaluates across a broad set of methods and models
4. Includes additional analysis (probability, ICL, concept-level) beyond raw metrics

Weaknesses:
1. Reverse-direction evaluation may be inherently more ambiguous or difficult, confounding conclusions
2. Lack of pre-edit baseline analysis to isolate editing-specific effects
3. Benchmark design (BAKE) is partly based on template reversal, with limited robustness analysis
4. Some claims about “lack of understanding” may be overstated

**Additional Comments:**

This paper identifies a meaningful and previously underexplored limitation of current model editing methods. The empirical results are strong and likely to influence how future work evaluates editing quality.

The main limitation lies not in the experiments themselves, but in the interpretation and evaluation design. In particular, the paper would benefit from more careful disentanglement between:
- intrinsic asymmetry in language models
- artifacts of reverse query formulation
- limitations of editing methods

Addressing these points would significantly strengthen the contribution and make the conclusions more robust.

**Audience:**

Yes

**Audience Explanation:**

The findings are highly relevant to researchers working on:
- Model editing
- Factual knowledge in LLMs
- Interpretability and controllability of language models

The paper highlights a previously overlooked failure mode—lack of bidirectional consistency—which has implications for:
- Reliability of edited models
- Safety and correctness of deployed systems
- Future design of editing algorithms

Even beyond model editing, the results contribute to a broader understanding of how LLMs encode and retrieve relational knowledge, which is of general interest to the TMLR community.

**Broader Impact Concerns:**

The paper already includes a discussion of potential misuse, particularly the risk that improved model editing could be used to reinforce misinformation. This is appropriate and relevant.

One additional point worth emphasizing is that:
- The findings suggest current editing methods may not reliably update knowledge in a consistent way
- This could lead to false confidence in edited models, especially in safety-critical applications

Overall, the broader impact discussion is adequate, and no major additional concerns are required.

**Claims And Evidence:**

Yes

**Claims Explanation:**

The paper provides extensive empirical evidence across multiple models, datasets, and editing methods. The reported performance gap between forward and reverse evaluation is large and consistent, which strongly supports the main empirical claim that current editing methods struggle in the reverse direction.

The analysis is also supported by additional experiments, such as probability comparisons showing that editing methods do not increase the likelihood of reverse answers, and ablations involving in-context learning.

However, while the empirical observations are convincing, the interpretation of these results is less conclusive. In particular:
- Reverse-direction queries may be inherently more difficult or ambiguous than forward queries
- The paper does not sufficiently compare against pre-edit baseline behavior to isolate whether the issue is caused by editing or is already present in the base model
- The metric does not normalize for task difficulty or relation type

Therefore, while the evidence for the phenomenon is strong, the causal interpretation (i.e., attributing it fully to editing failure) is somewhat less rigorously supported.

**Requested Changes:**

1. Control for inherent difficulty of reverse queries
- Provide analysis comparing reverse-direction performance before and after editing
- Clarify whether the observed gap is due to editing or intrinsic asymmetry in LLMs
2. Better justification of benchmark validity
- More clearly discuss how BAKE controls for ambiguity in reverse relations
- Provide breakdown by relation type (e.g., one-to-one vs one-to-many) with deeper analysis
3. Metric calibration
- Consider normalizing reversibility scores with respect to:
    - base model performance
    - relation difficulty
- Alternatively, report relative changes instead of absolute scores

---

> ### Author Response · Authors · 2026-03-30
>
> ## **Re Weakness 1: Reverse-direction evaluation may be inherently more ambiguous or difficult, confounding conclusions**
>
> We thank the reviewer for the insightful comment regarding the potential difficulty or ambiguity of reverse queries.
>
> We agree that reverse queries may, in some cases, be inherently more ambiguous or challenging. In fact, one of the motivations of this work is precisely to highlight that reverse reasoning remains a **challenging problem for current language models**.
>
> However, our goal is not to measure the **absolute difficulty of reverse queries**, but to examine whether editing operations can successfully propagate knowledge to the reverse direction under this challenging setting.
>
> Under this setup, we observe a consistent pattern: **forward editing performance improves significantly after editing, while reverse performance remains substantially lower**. This systematic asymmetry suggests that, even when reverse queries are inherently difficult, existing editing methods fail to effectively model inverse relations and propagate knowledge bidirectionally.
>
> Therefore, the observed performance gap cannot be fully explained by the difficulty of reverse queries alone. Instead, it highlights a **fundamental limitation of current editing methods in achieving bidirectional generalization**, which is precisely the core problem this work aims to reveal.
>
> ## **Re Weakness 2: Lack of pre-edit baseline analysis to isolate editing-specific effects**
>
> We thank the reviewer for this important comment regarding the lack of pre-edit baseline comparison.
>
> We would like to clarify that our experimental design already **explicitly controls for pre-edit knowledge**. As described in Section 4.2, we filter out all counterfactual editing triples that already exist in the base model. As a result, the pre-edit reverse performance (RQS and RJS) is **consistently zero across all models** (as reported in Section 5.2).
>
> This design ensures that the reverse-direction evaluation is not influenced by pre-existing knowledge, allowing us to clearly attribute performance changes to the effect of editing operations. From this perspective, our evaluation can be viewed as **implicitly measuring relative improvements**, since all models start from a near-zero baseline in the reverse direction.
>
> Under this controlled setup, we observe that **editing significantly improves forward performance, while the improvement in reverse performance remains limited**. This asymmetric behavior further indicates that existing editing methods fail to propagate knowledge to the reverse direction.
>
> Therefore, we believe that our experimental design effectively accounts for pre-edit baselines and supports attributing the observed performance gap to **structural limitations of current editing methods in achieving bidirectional generalization**.
>
> ## **Re Weakness 3: Benchmark design (BAKE) is partly based on template reversal, with limited robustness analysis**
>
> We thank the reviewer for the insightful comments regarding benchmark design, template-based construction, and potential ambiguity.
>
> We would like to clarify that **ambiguity and potential bias are explicitly considered and controlled** in the design of BAKE.
>
> First, at the relation level, we categorize relations into four types: one-to-one, one-to-many, many-to-one, and many-to-many (as described in Section 4.2), and adopt different evaluation strategies accordingly. For one-to-one and one-to-many relations, where the inverse relation has a unique answer, we use both QA and judgment formats. In contrast, for many-to-one and many-to-many relations, where reverse queries may admit multiple valid answers, we do not use QA evaluation, but instead rely solely on the judgment format. This design **avoids ambiguity in inherently multi-answer settings**.
>
> Second, regarding template construction, BAKE is **not based on naive string-level template reversal**. Instead, reverse queries are constructed based on the semantic roles of relations (subject–relation–object). In addition, we incorporate paraphrase prompts and multiple evaluation formats to reduce the likelihood that models rely on superficial pattern matching.
>
> More importantly, our key observation, **significant improvement in forward performance but consistently limited reverse performance**, holds across multiple models, relation types, and evaluation settings. This suggests that the observed phenomenon is not an artifact of template design or ambiguity, but instead reflects a general limitation of current editing methods in modeling inverse relations.
>
> Therefore, we believe that BAKE incorporates explicit mechanisms to control ambiguity and avoids the pitfalls of naive template reversal, enabling a **robust evaluation of bidirectional reasoning** in model editing.

---

> > ### Author Response · Authors · 2026-03-30
> >
> > ## **Re Weakness 4: Some claims about “lack of understanding” may be overstated**
> >
> > We thank the reviewer for the valuable comment regarding potentially overstated claims.
> >
> > We agree that some expressions in the original manuscript (e.g., “lack of understanding”) may be **overly strong** and could lead to unintended interpretations. In the revised version, we have carefully reviewed and refined these statements, adopting **more precise and cautious wording** to improve clarity and avoid overstatement.
> >
> > At the same time, we emphasize that these revisions concern the **phrasing rather than the underlying findings**. Our core observation, which is existing editing methods exhibit substantial limitations under reverse inference, remains well supported by the experimental results.
> >
> > ## **Re Requested Changes**
> >
> > We sincerely thank the reviewer for the comprehensive and constructive suggestions provided in the Requested Changes.
> >
> > Regarding the control of reverse query difficulty, we have clarified that our experimental design explicitly filters out pre-existing knowledge, resulting in near-zero pre-edit reverse performance. This allows us to **isolate the effect of editing** and attribute the observed asymmetry to the editing process rather than intrinsic difficulty alone.
> >
> > For benchmark validity, we have further explained that BAKE explicitly accounts for relation structure by categorizing relations into four types and adopting **relation-aware evaluation strategies**. In particular, we avoid ambiguity in multi-answer settings by using judgment-based evaluation instead of QA for many-to-one and many-to-many relations.
> >
> > Regarding metric calibration, we note that our evaluation effectively operates on a **relative scale**, as all models start from a near-zero baseline in the reverse direction. This implicitly captures relative improvements, and we will further clarify this perspective in the revised manuscript to better align with the suggested normalization strategies.
> >
> > Overall, we believe that these clarifications address the reviewer’s concerns and further strengthen the validity and interpretability of our evaluation framework.
> >
> > These changes improve the overall **rigor and objectivity** of the presentation.

---

> > > ### Author Response · Authors · 2026-04-04
> > >
> > > Dear Reviewer,
> > >
> > > We sincerely appreciate your time and the insightful feedback you provided. We hope our rebuttal effectively addresses your concerns. Please let us know if you have any remaining questions so we can provide further clarification in time.
> > >
> > > Authors

---

### Author Response · Authors · 2026-03-29

Dear Action Editor,

I am writing to kindly follow up on the status of the third review for our paper. I just wanted to gently check in, as there appears to have been some delay with the third review. We truly appreciate the time and effort involved in coordinating the reviews.

Thank you very much for your time and help.

Authors

---

### Decision · Action_Editor_ZuoA · 2026-05-25

**Recommendation:** Accept with minor revision

**Additional Comments:**

This paper studies whether the edited model can correctly answer the question in the reverse direction. The authors have constructed BAKE, a benchmark to evaluate model's reverse generalization capability. The authors have further provided several empirical analysis and lessons learned based on extensive experiments conducted on BAKE, including the output probability in the reverse direction and the impact of in-context learning for model editing.

Reviewers overall acknowledge the contributions of this paper. The authors have done substantial amount of experiments in the original manuscript and as suggested by the reviewers. After checking the paper, reviewer comments, and author-reviewer discussions, I recommend acceptance with minor revisions. The authors are suggested to add more discussions regarding the ambiguity/difficulty of reverse direction and the robustness with respect to template generation, as well as any promised changes in the updated version.

**Audience:**

Yes

**Audience Explanation:**

Model editing for LLMs has been an important topic, due to the importance of timeliness of LLM world knowledge. I believe this topic would be of interest to TMLR audience who are working on LLM-related topics.

**Claims And Evidence:**

Yes

**Claims Explanation:**

The claims made in the submissions are supported by extensive amount of empirical experiments in the manuscript.